# LANA-specific CD4+ effector T cells accumulate at the site of KSHV infection in humanized mice

Michelle Böni [1], Shitao Peng[1], Danusia Vanoaica[1], Kareem Haal [1,3], Svenja L. Nopper[1], Lisa Rieble [1,4], Sandra Schmid[1], Alma Delia Valencia-Camargo[1,5], Angelika Holler[2], Hans Stauss [2] & Christian Münz [1] ✉

Kaposi's sarcoma-associated herpesvirus (KSHV) infection is linked with the development of life-threatening malignancies in elderly and immunocompromised hosts, suggesting tight control of the infection by T cell responses. T cells against KSHV, however, are barely detectable in infected individuals, and the mechanisms underlying immune recognition of KSHV-infected cells remain poorly understood. Here, we present publicly available sequences of T cell receptors (TCRs) targeting the KSHV latency-associated nuclear antigen (LANA/ORF73). By using these TCRs transgenically expressed on T cells as identifiers for KSHV-specific cells, we show that despite their failure to recognize KSHV-infected B cells in vitro, activated effector memory differentiated LANA-specific CD4+ T cells accumulate in vivo at infection sites in the preclinical infection model of humanized mice. This suggests more efficient antigen-presentation in vivo than by KSHV-infected B cells in vitro and highlights the possible contribution of CD4+ T cells to the immunosurveillance of latently infected B cells.

Kaposi's sarcoma-associated herpesvirus (KSHV, HHV8) and Epstein-Barr virus (EBV, HHV4) are human ɣ-herpesviruses characterized by their ability to establish latent infections of B cells, endothelial and epithelial cells[1]. The coevolution of the phylogenetically old ɣ-herpesviruses with the human host resulted in finely balanced interactions with the host immune system, enabling high population penetration with seroprevalence rates exceeding 90% depending on the geographic region[1,2]. However, disruptions in the delicate host-pathogen equilibrium, particularly within the T cell compartment, can result in immunopathology and cancer, thus explaining the association of EBV and KSHV with around 1–2% of malignancies[3,4].

EBV is a potent driver of T cell responses, with EBV-specificities constituting up to 50% of total CD8+ T cells during symptomatic primary EBV infection[5–7]. Primary immunodeficiencies affecting the cytotoxic machinery (perforin), leucocyte development (GATA2), TCR signaling (ZAP70, RasGRP1), or co-stimulatory pathways (CD27, 4-1BB, XLP1) ultimately impair T cell function and predispose individuals to EBV-associated diseases[8]. Furthermore, preclinical T cell depletion experiments as well as the curative adoptive transfer of EBV-specific CD8+ T cells in lymphoma and posttransplant lymphoproliferative disorder patients underscore the critical role of T cells in controlling EBV infection[9–12].

[1]Viral Immunobiology, Institute of Experimental Immunology, University of Zurich, Zurich, Switzerland. [2]Institute of Immunity and Transplantation, Division of Infection and Immunity, University College London, Pears Building, London, UK. [3]Present address: Department of Gastroenterology, Hepatology, Infectious Diseases and Endocrinology, Hannover Medical School, Hannover, Germany. [4]Present address: Department of Medicine Huddinge, Center for Infectious Medicine, Karolinska Institutet, Stockholm, Sweden. [5]Present address: Institute of Pharmaceutical Sciences, ETH Zurich, Zurich, Switzerland. ✉e-mail: christian.muenz@uzh.ch

In contrast, T cell-mediated control of KSHV is less well characterized, but immunodeficiencies affecting TCR signaling (STIM1, WASP), co-stimulation (OX40, XMEN) or T cell effector functions (STAT4, IFNγR1) are also known to predispose for Kaposi sarcoma (KS) development, the most common malignancy associated with KSHV[8]. These mutations suggest that IFNγ production (STAT4, IFNγR1) and possibly CD4+ T cells (OX40) might play a larger role in the immune control of KSHV than EBV. Accordingly, no massive CD8+ T cell expansions, which are characteristic of symptomatic primary EBV infection, have been observed during KSHV seroconversion in HIV-negative individuals[13,14]. However, the occurrence of KSHV-specific IFNγ responses, alongside KSHV viremia, suggests the activation and priming of virus-specific T cells[14]. Nevertheless, KSHV-specific T cell reactivities per individual appear to be limited, with patients recognizing on average only 1–5 different KSHV antigens, regardless of immune status, compared to a mean of 21 different EBV antigens[15]. Moreover, these KSHV-specific responses are highly heterogenous in their viral antigen recognition between individuals[16,17]. The lack of knowledge regarding conserved KSHV antigens presented by specific major histocompatibility complex (MHC) molecules poses challenges for broad tetramer-based assays or the identification of KSHV-specific T cells. Nonetheless, among the most frequently targeted KSHV antigens are K8.1 and LANA (ORF73), with approximately half of patients showing T cell responses against at least one of these[15]. K8.1 is a glycoprotein likely recognized during primary infection, whereas LANA is consistently expressed in latently infected cells, where it tethers the viral episome to host chromosomes[18–20]. Notably, both proteins are also prominent targets of antibody responses to KSHV[21–23]. However, responses detected against these antigens are weak, and LANA-specific CD4+ T cells demonstrated poor recognition of infected B cells in vitro[15–17,24–26].

Despite these low frequencies of KSHV-specific T cell responses, epidemiological data strongly argue for a critical role of robust and effective T cell surveillance in controlling KSHV. KS, primary effusion lymphoma (PEL), multicentric Castleman's disease (MCD) and KSHV inflammatory cytokine syndrome, all KSHV-associated diseases, are most prevalent in patients with HIV infection, under iatrogenic immunosuppression or in elderly males with waning immunity[27]. Improved outcomes associated with combined antiretroviral therapy, reduced immunosuppression and promising clinical results of immune checkpoint blockade in KS further highlight the importance of T cells in disease control[28–30]. Additionally, T cell depletion experiments in EBV and KSHV-co-infected mice with reconstituted human immune systems (humanized mice) have demonstrated a protective role for T cells[31].

Humanized mice co-infected with EBV have emerged as a small animal model for studying persistent KSHV-infection in B cells[31–33]. Evidence from in vitro studies, epidemiological cohort data and the characteristic presence of EBV in 90% of PEL cases suggests that EBV may facilitate KSHV infection and persistence in B cells[1,34–37]. Only PEL, but not KS or MCD tumor cells, harbor EBV co-infection. In line with this, PEL-like B cell tumors have been detected in EBV- and KSHV-co-infected humanized mice. These mice have further demonstrated the ability to generate KSHV-specific T cell and IgM responses[31–33]. However, KSHV-specific T cells have not yet been characterized in vivo throughout the course of infection.

In the present study, we aimed to use T cell receptors (TCRs) specific for LANA as markers to identify KSHV-specific T cells. We sequenced and cloned CD4+ and CD8+ T cell-derived LANA-specific TCRs and made these publicly available for future research. We show that despite poor recognition of KSHV-infected B cells by LANA-specific TCR-transduced T cells in vitro, CD4+ LANA-specific effector memory T cells (TEM) accumulate at infection sites in vivo and acquire an early differentiated TEM phenotype.

## Results

### Generation of transgenic KSHV-specific T cell receptors

To generate transgenic KSHV-specific TCRs, we made use of previously described primary T cell clones kindly provided by Dr. Jianmin Zuo[24]. T cells were originally selected from peripheral blood mononuclear cells (PBMCs) of KSHV-infected human individuals by overlapping peptide pools of LANA and subsequently single-cell cloned. All but one clone (Cl12) were CD4+ T cells. We started by confirming the described antigen specificity via a marked upregulation of CD107a upon co-culture of the CD4+ T cells with cognate peptide-pulsed autologous lymphoblastoid cell lines (LCLs). For the CD8+ T cell clone (Cl12), we used peptide-pulsed T2 cells (a fusion product of LCL721.174 and CEM cells that mainly retained LCL721.174 characteristics) transfected with HLA-B*35:01 (T2B35 cells[38,39],) (Fig. 1A). We further assessed the clonality of the T cells by flow cytometric staining for the TCR variable beta chain (Vß) repertoire. For non-clonal populations, this approach, combined with the CD107a degranulation assay, enabled identification of the TCR Vß of the CD107a+ reactive subpopulation (Fig. S1A). After sequencing the TCRs of all clones, the variable chains were codon-optimized and cloned into a modified TCR construct of a lentiviral vector[40]. This construct contains mouse constant TCR-chain domains along with additional disulfide bonds to enhance pairing of the transgenic TCR α and ß chain (Fig. 1B). It allows for detection of the transgenic TCR on the cell surface via staining for the mouse TCR constant beta chain (mCB), as well as general detection of transduced cells through staining for a truncated version of mouse CD19 (mCD19, Figs. 1B, and S1B). Expression of the TCR was assessed on transduced Jurkat Lucia NFAT reporter cells (Fig. S2B), and functionality of the transgenic TCR was assessed by coculturing the transduced reporter cells with cognate peptide-pulsed HLA-matching target cells. The reporter cells, which encode an NFAT-inducible Lucia luciferase, facilitated the detection of TCR engagement and signaling via luminescent measurement of Lucia luciferase activity. For all clones except Cl55, we successfully established functional KSHV-specific TCRs (Fig. 1C). Cl33 and Cl87 shared identical TCR sequences and are hereafter referred to collectively as Cl33. None of the identified CDR3 amino acid sequences matched curated antigen specificities in the VDJdb[41,42] or in the McPAS-TCR databases[43] and was therefore not associated with known specificity for other pathogens. Functional transgenic TCRs were further characterized for their HLA-restriction (Figs. 1D and S6A). Transduced Jurkat Lucia NFAT cells were cocultured with partially HLA class II-matched, cognate peptide-pulsed LCLs. All CD4+ T cell-derived TCRs were HLA-DRB1*13:02 restricted, and they were able to interact to similar levels with certain allotypes of HLA-DRB1*13:02 (HLA-DRB1*13:01, HLA-DRB1*13:103, and HLA-DRB1*13:11 (Cl33), but not HLA-DRB1*13:03 and HLA-DRB1*13:11 (Cl156)). The CD8+ T cell-derived TCR, in contrast, demonstrated strict specificity for HLA-B*35:01 and did not interact with the related allotypes HLA-B*35:08 and HLA-B*35:32, as later determined through cognate peptide-specific IFNγ production from primary transduced T cells (Figs. 1D and S1C). Overall, we cloned three KSHV LANA-specific HLA-DRB1*13 restricted TCRs and one KSHV LANA-specific HLA-B*35:01 restricted TCR.

### LANA-specific TCR-transduced T cells demonstrate cytotoxic potential

We next assessed the expression and functionality of the KSHV-specific TCRs on primary T cells. CD4+ or CD8+ T cells were isolated from PBMCs of healthy donors and lentivirally transduced with the respective TCRs (Figs. 2A and S2A). Primary T cells exhibited consistently lower transduction and TCR expression rates compared to Jurkat Lucia NFAT cells, but maximum levels were reached at similar virus amounts, with no further increase in transduction or TCR expression upon adding more virus (Fig. S2B, C). For each TCR, we measured the functional avidity via exposure to HLA-matched LCLs pulsed with decreasing amounts of cognate peptide. In primary T cells, avidity was

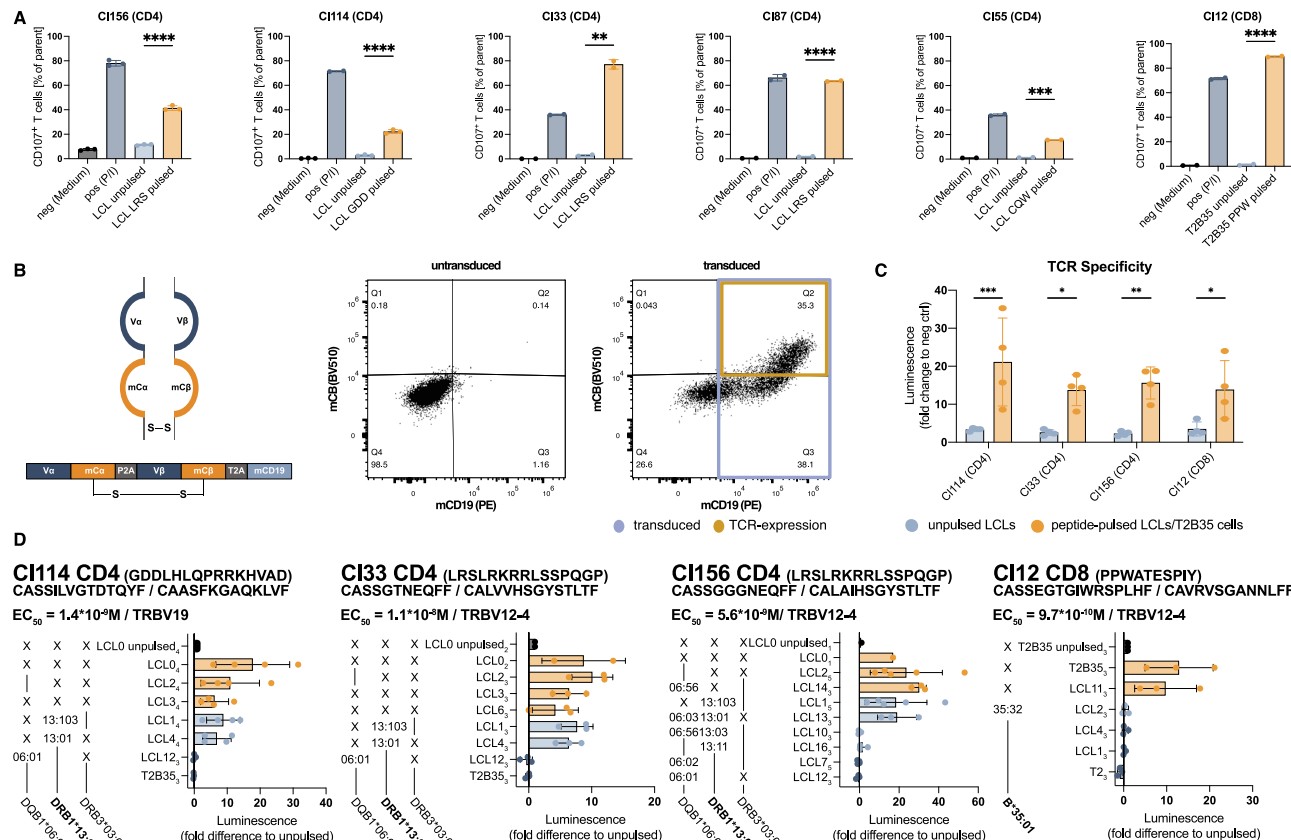

**Fig. 1 | Generation of transgenic KSHV-specific T cell receptors from LANA-specific T cell clones. A** Degranulation assay measuring the frequency of CD107a+ cells of CD4+ or CD8+ T cell clones upon co-culture with medium only (neg), PMA and Ionomycin (pos P/I), autologous unpulsed or cognate peptide pulsed LCLs or T2B35 cells. Mean ± SD of duplicates (Cl33/87/55/12/114pos) or triplicates (Cl156/114) of n = 1 experiment. Two-tailed unpaired t-test comparing unpulsed and pulsed conditions. **B** Schematic representation of the TCR construct used: variable α chain (Vα) followed by mouse constant α chain (mCα), P2A, variable ß chain (Vß), mouse constant ß chain (mCß), T2A, truncated mouse CD19 (mCD19). mCα and mCß are linked by an additional disulfide bond. Representative flow cytometry scatter plots of untransduced and transduced Jurkat Lucia NFAT cells showing the gating for transduced and TCR-expressing cells. **C** Jurkat Lucia NFAT reporter assay showing the fold change luminescent signal normalized to the background of unstimulated cells for transgenic TCR-transduced Jurkat Lucia NFAT cells stimulated with HLA-matched pulsed or unpulsed LCLs or T2B35 cells.

Graph shows mean ± SD of n = 4 independent experiments. Two-tailed unpaired t-test. **D** Overview of the characteristics of each cloned TCR. Title contains information on the cognate antigenic peptide in brackets, the CDR3α and CDR3ß amino acid sequences, the TCR-Vß usage as well as the functional avidity EC50 as determined in Fig. 2. Plot shows the HLA-restriction assessed via Jurkat Lucia NFAT reporter assays: fold difference of normalized luminescence signals of pulsed versus unpulsed LCLs or T2B35 cells is plotted for target cells sharing the same HLA-allotype (orange, x), target cells sharing the same HLA-allele group (light blue, HLA-allele group variants) and HLA-mismatched target cells (dark blue, line). The autologous LCL0 from the CD4+ T cell donor is DQB1*06:09, DRB1*13:02 and DRB3*03:01 positive. Graph shows mean ± SD of n = 1-5 independent experiments. Exact number n for each target cell tested is indicated subscripted in the y-axis labels. (**A–D**) *p < 0.05, **p < 0.01, ***p < 0.001, ****p < 0.0001. Exact significant p ≥ 0.0001 from left to right (1A) p = 0.0014, p = 0.0001; (1C) p = 0.0009, p = 0.0282, p = 0.0089, p = 0.0427.

assessed by IFNɣ detection in the supernatant and in Jurkat Lucia NFAT reporter cells by Luciferase activity measurement (Fig. 2B). We observed a left shift in the response curve of primary T cells (higher avidity) compared to the Jurkat Lucia NFAT cells, which may be attributed to the lack of the CD8 co-receptor, the significantly reduced CD4 expression (Fig. S2D), and the lack of CD28 expression (providing co-stimulatory signal 2) on Jurkat Lucia NFAT cells. The calculated EC50 values of the transgenic primary T cells ranging from $10^{-8}$ to $10^{-10}$ M were comparable to the functional avidities originally reported for the primary T cell clones that endogenously expressed the KSHV-specific TCRs[24]. Apart from producing IFNɣ upon co-culture with cognate peptide-pulsed HLA-matched LCLs (Fig. 2C), both CD4+ and CD8+ T cells seemed to be able to activate and recruit the cytotoxic machinery: We observed a significantly increased expression of CD107a in the peptide-pulsed compared to the unpulsed co-cultures (Figs. 2D, F and S3A) and cognate-peptide pulsed target cells were killed more readily by both, CD4+ and CD8+ T cells (Figs. 2E and S3B). This is supported by an increased frequency of Granzyme B expressing CD4+ and CD8+ T cells upon TCR-activation (Figs. 2F, G and S3A). This is

in line with previously reported cytotoxicity of the KSHV-specific CD4+ T cell clone as assessed by $^{51}$Cr-release assay[24]. Additionally, we observed an increased frequency of cells expressing TNF and IFNɣ, and an increase in cells expressing the activation markers PD1, HLA-DR, OX40 and CD137 upon activation (Figs. 2F, G and S3A, C). CD40L was not measurably upregulated, probably due to the lack of the addition of a CD40 blocking antibody. In summary, TCR-transduced transgenic KSHV-specific T cells recapitulate features of endogenous KSHV-specific T cells in vitro[24] and are therefore a valuable tool to study KSHV-specific T cell responses.

**LANA-specific T cells fail to recognize KSHV-infected B cells**

T cells, including the LANA-specific T cells clones whose TCRs were used in this study, have been shown to poorly recognize naturally infected PEL cells or KSHV-single-infected tonsillar B lymphocytes[24,25,44,45]. Furthermore, Cl33 TCR-transduced T cells failed to recognize the HLA-matched HLA-DRB1*13:01 positive VG-1 PEL cell line unless peptide pulsed (Fig. S4A). Yet, about 90% of PEL cases are co-infected with EBV, and recent studies suggest that EBV gene

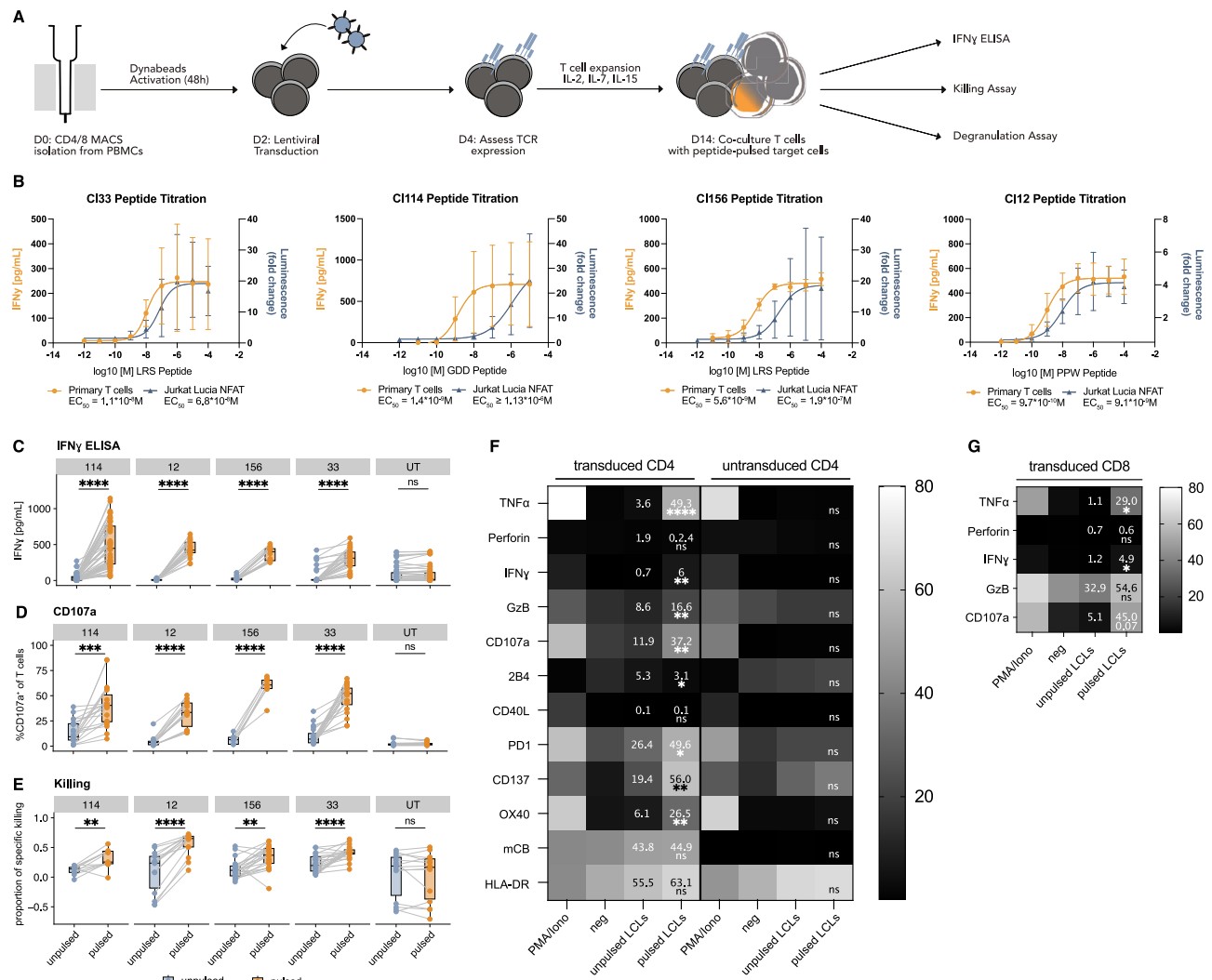

**Fig. 2 | LANA-specific TCR-transduced CD4+ and CD8+ T cells demonstrate cytotoxic potential in vitro. A** Schematic of primary TCR-transduced T cell generation for downstream assays. **B** IFNɣ levels of primary TCR-transduced T cells or normalized luminescence of TCR-transduced Jurkat Lucia NFAT cells co-cultured overnight with HLA-matched LCLs pulsed with decreasing peptide concentrations. Graph shows mean ± SD of $n = 2$ (Jurkat Cl33), $n = 3$ (Jurkat Cl156/12, primary Cl114) or $n = 4$ (rest) independent experiments. **C** IFNɣ ELISA of TCR-transduced T cells co-cultured with cognate peptide-pulsed or unpulsed HLA-matched LCLs. Boxplots represent 5 (Cl114, Cl33), 4 (Cl12) or 3 (Cl156) independent experiments, using T cells from 8 (Cl114, Cl33, Cl12) or 6 (Cl156) different donors, tested against 9 (Cl114), 6 (Cl33), 4 (Cl156), 3 (Cl12) different LCLs. **D** Frequency of CD107a+ T cells after 6 h of co-culture with cognate peptide-pulsed or unpulsed HLA-matching LCLs. Boxplots represent 3 (Cl114, Cl33, Cl12) or 1 (Cl156) independent experiments, using T cells from 6 (Cl12), 4 (Cl114, Cl33) or 2 (Cl156) different donors, tested against 6 (Cl114), 5 (Cl33), 4 (Cl156) or 3 (Cl12) different LCLs. **E** Specific killing of

HLA-matched pulsed or unpulsed LCLs after co-culture with TCR-transduced T cells. Boxplots represent 3 (Cl156, Cl33, Cl12) or 2 (Cl114) independent experiments, using T cells from 6 (Cl156, Cl33), 5 (Cl12) or 3 (Cl114) different donors, tested against 5 (Cl114, Cl156, Cl33) or 2 (Cl12) different LCLs. **C–E** Paired two-tailed t-test. Intracellular cytokine staining (6 h) and surface marker assessment (18 h) of CD4+ (**F**) and CD8+ (**G**) T cells after stimulation with PMA/Ionomycin, cognate peptide-pulsed or unpulsed LCLs or medium only as a negative control. Mean frequency of cells positive for the indicated marker (y-axis) in each condition (x-axis) is plotted from 3 (CD4+) or 1 (CD8+) independent experiments with 5 (CD4+) or 2 (CD8+) different T cell donors. Numerical means are shown for LCL-stimulated conditions. Asterisks indicate significance from paired two-sided t-tests comparing pulsed versus unpulsed LCLs. **A–E** *$p < 0.05$, **$p < 0.01$, ***$p < 0.001$, ****$p < 0.0001$. Exact significant $p \geq 0.0001$ from left to right (2D) 0.00015; (2E) $p = 0.00236$, $p = 0.00019$.

expression is critical for KSHV persistence in B cells in vitro and in vivo not only in PEL cells[1,33,35,46–48]. Consequently, infection of primary B cells with EBV and KSHV results in stably transformed B cells, providing a new model to study immune recognition of KSHV in the context of EBV co-infection using freshly infected B cells[31,35,48]. We therefore established co-infected B cells (EK LCLs) from different HLA-matched donors and tested their immunogenicity in an IFNɣ assay. As LANA-specific, TCR-transduced T cells failed to recognize these cells (Fig. S4B), we enriched the population of KSHV-infected cells to virtually 100% by selection with Puromycin (Fig. 3A). The presence of KSHV was confirmed via flow cytometric detection of GFP constitutively expressed from the recombinant rKSHV.219 (Fig. 3B).

Additionally, expression of LANA, the target of our transgenic TCR, was verified by immunoblotting (Fig. 3C). Nevertheless, TCR-transduced LANA-specific T cells also failed to recognize KSHV-enriched LCLs: T cell effector functions, such as IFNɣ or TNF production, target cell killing, or T cell degranulation were not increased when stimulated with KSHV-infected LCLs compared to EBV-only LCLs (Figs. 3D and S4C, D, S5A). Only for Cl114, stimulation with EK LCLs resulted in an average 1.1-fold higher IFNɣ secretion, but this significant difference was mainly attributed to four outliers in the experimental dataset (Fig. 3D). Similarly, no enhanced proliferation of LANA-specific T cells was observed after seven days of co-culture with EK LCLs, as indicated by unchanged frequencies of CellTrace Violet diluted cells or

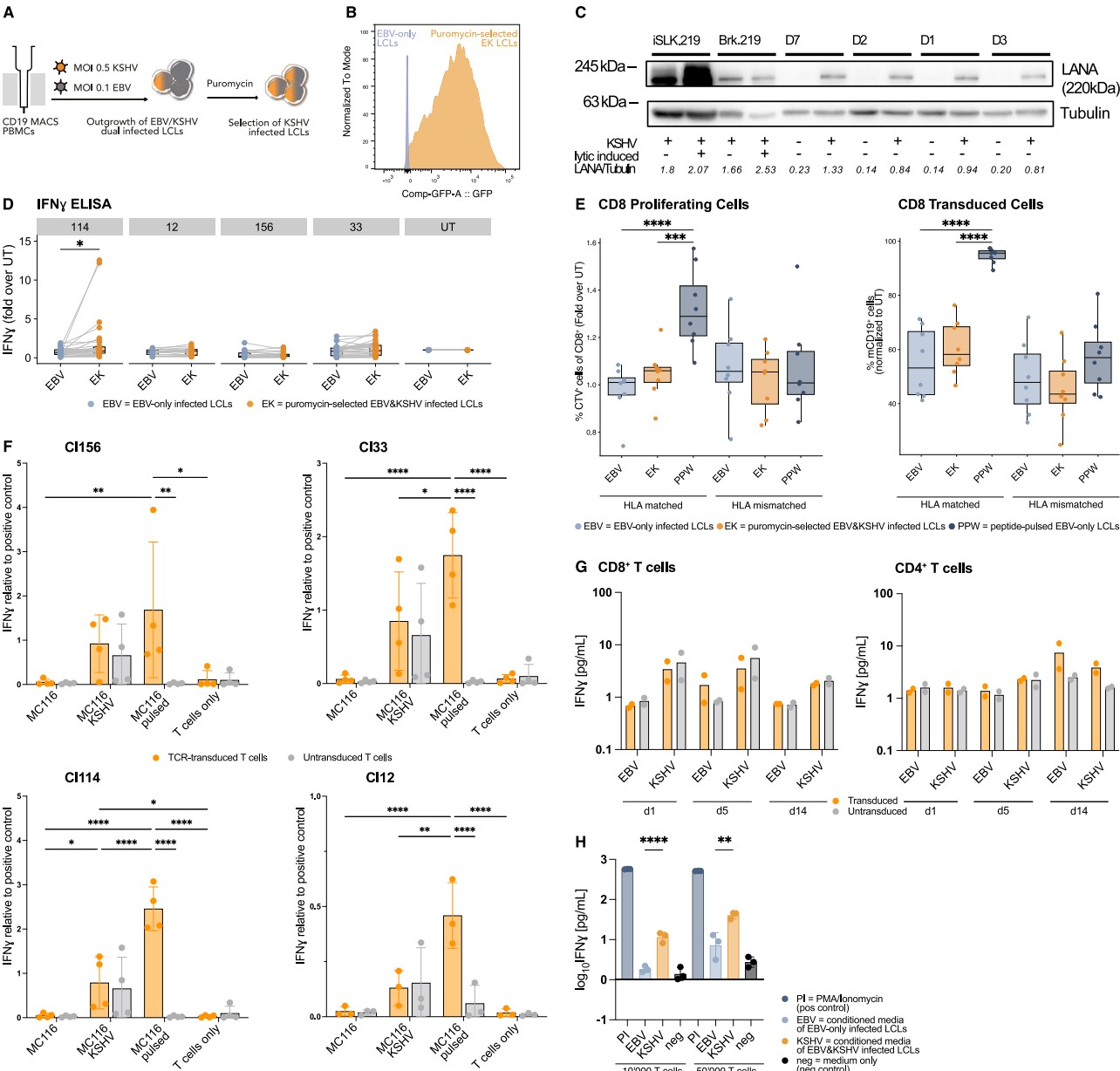

**Fig. 3 | LANA-specific CD4+ and CD8+ T cells fail to recognize KSHV-infected B cells through a TCR-dependent mechanism. A** Schematic of EBV&KSHV co-infected LCL (EK LCL) generation. **B** Flow cytometry histograms of GFP expression in different LCLs. **C** Western blot of whole-cell lysates from iSLK.219 and Brk.219 (± lytic induction) and from EBV-only or puromycin-selected EK LCLs (4 donors) stained for LANA and Tubulin. Tubulin-normalized LANA quantification indicated below blots. **D** IFNγ production by TCR-transduced T cells co-cultured with HLA-matched LCLs (± KSHV infection), normalized to untransduced controls. Boxplots representing five (Cl33), four (Cl114, Cl12) or three (Cl156) independent experiments using T cells from eight (Cl114, Cl33, Cl12) or six (Cl156) donors, tested against six (Cl114, Cl33), four (Cl156) or three (Cl12) LCLs. Paired two-tailed t-test. **E** Frequencies of proliferating (CTV-diluted) or mouse CD19+ CD8+ T cells after seven-day co-culture of Cl12-transduced CD8+ T cells with irradiated HLA-matched or mismatched LCLs (± KSHV infection, ± peptide-pulsed), normalized to untransduced T cells. Boxplots representing two independent experiments with four T-cell

and two target-cell donors. Linear mixed model fit by REML, T cell and LCL donor as a random effect, Holm-adjusted Tukey test. **F** IFNγ secretion by T cells co-cultured with peptide-pulsed, KSHV-infected, or untreated MC116 B cells, or medium only. IFNγ levels are shown relative to PMA/Ionomycin controls. Median ± SD of 2 independent experiments using T cells from three (Cl12) or four (rest) donors. Two-way ANOVA, Tukey's multiple comparisons test, multiplicity adjusted p-values. **G** IFNγ ELISA of TCR-transduced T cells co-cultured with autologous B cells infected with EBV or EBV&KSHV on days 1, 5, 14 post-infection. Mean of one experiment with two donor pairs. **H** IFNγ ELISA of untransduced T cells incubated 72 h with conditioned media from freshly infected B cells. Mean ± SD of one experiment with three donor pairs. One-way ANOVA on log10-transformed data, Tukey's multiple comparisons test. (**A**–**H**) *p < 0.05, **p < 0.01, ***p < 0.001, ****p < 0.0001. Exact significant p ≥ 0.0001 (left to right, top to bottom): (3D) p = 0.03852; (3E) p = 0.00034; (3 F) Cl156: p = 0.0078, p = 0.0013, p = 0.0107; Cl33: p = 0.0224; Cl114: p = 0.0402, p = 0.0472; Cl12: p = 0.0016; (3H) p = 0.0037.

unchanged frequencies of the transgenic cells within the T cell population (Figs. 3E and S5C). For CD4+ T cells, proliferation and transgenic cell frequencies also showed no increase upon stimulation with peptide-pulsed LCLs, likely due to already high baseline levels (Fig. S5B). However, looking at the EK LCLs as target cells, MHC class I

and II expression levels were comparable to those observed in EBV-only infected LCLs, but CD86, ICAM-1 and CD40 were markedly downregulated (Fig. S5D). These immunomodulatory features are in part consistent with those observed on circulating KSHV-infected vir-oblasts during MCD, KSHV-infected endothelial cells and PEL cell

lines[44,49,50]. Yet, the failure to react against EK LCLs is likely due to an insufficient TCR signal, as also Jurkat Lucia NFAT cell reporter activity, which solely depends on signal 1, showed no increase after co-culturing TCR-transgenic cells with EK LCLs compared to EBV-only LCLs (Fig. S5E). We further explored whether LANA might be recognized in alternative host cell states, such as recently infected or lytic cells. Although less studied for KSHV, EBV induces significant transcriptional changes in host cells, particularly within the first 14 days of infection[51–53]. To investigate this, T cell recognition of the MC116 B cell line 48 h post-infection and of Brk.219 cells 48 h post-lytic induction, were assessed. KSHV infection of the HLA-B*35:01+ HLA-DRB1*13:01+ MC116 cell line indeed induced increased IFNγ secretion by T cells, but independent of the transgenic TCR expression (Figs. 3F and S6A). A similar pattern was observed in lytically induced B*35:01+ HLA-DRB1*13:02+ Brk.219 cells (Fig. S6B). Additionally, we co-cultured TCR transgenic T cells with autologous CD19+ B cells on day 1, 5 and 14 post-infection with EBV or EBV plus KSHV. During these first 2 weeks of infection, no TCR-dependent recognition of KSHV-infected cells was observed, though IFNγ secretion increased again slightly upon stimulation with KSHV-infected cells. (Fig. 3G). This subtle increase in effector function, observed after stimulation with both, recently KSHV-infected cell lines and EK LCLs (Figs. 3F, G, and S6C), is likely mediated by paracrine signaling, as supernatant transfer from infected cells was sufficient to induce increased IFNγ secretion (Figs. 3H and S6D). Overall, EK LCLs do not seem to be more immunogenic, as peptide-pulsed EK LCLs elicited responses similar to peptide-pulsed EBV-only LCLs (Fig. S4C). In conclusion, no specific cell state was identified in which KSHV-infected B cells, as professional antigen-presenting cells (APCs), could stimulate a LANA-specific T cell response.

## LANA-specific CD4+ T cells accumulate in the peritoneal cavity of KSHV-infected humanized mice

KSHV-associated B cell malignancies, such as PEL, have the highest incidence in HIV and other immunocompromised patients, suggesting T cells to be critical for virus control[27]. Despite failing to recognize KSHV-infected B cells in vitro, we therefore continued to explore LANA-specific TCR-transgenic T cells during KSHV infection in vivo. Co-infection of humanized mice with EBV and KSHV has been shown to promote PEL-like tumorigenesis by driving lymphoproliferations composed of EBV and KSHV-co-infected B cells, while simultaneously inducing adaptive KSHV-specific immune responses, notably priming KSHV-specific T cells[31–33]. This makes it a suitable model for studying KSHV-specific T cells, using TCR-transgenic cells as indicators during primary infection. Cl33 TCR-transduced CD4+ T cells as well as Cl12 TCR-transduced CD8+ T cells were generated from splenocytes of humanized mice and intravenously injected into donor-mate animals reconstituted with the same CD34+ hematopoietic progenitor cells (HPCs, Fig. 4A). HPCs were HLA-B*35:01 and HLA-DRB1*13 positive and recognition of the peptide-MHC complex by Cl33 TCR was confirmed (Fig. S7A). One day after adoptive T cell transfer, animals were infected with EBV and KSHV and monitored for 4 weeks. KSHV-specific T cells were identified by flow cytometric staining for mCD19 and mCB (Figs. 4B and S7B). At the study endpoint, CD8+ transgenic T cells were primarily found in the spleen at frequencies below 1%, with occasional detection in the peritoneal lavage and bone marrow (Fig. 4D). In contrast, CD4+ transgenic T cells were detected in all analyzed organs at mean frequencies ≥0.1%, including spleen, bone marrow, blood and peritoneum, although they were absent from the bone marrow and blood in some animals (Fig. 4C). CD4+ and CD8+ transgenic T cells only transiently appeared in the blood, with no expansion over the course of the experiment. Unlike TCR transgenic CD8+ T cells, KSHV-specific CD4+ T cells preferentially homed to the peritoneal cavity, comprising up to 5% of total CD4+ T cells at that site (Fig. 4E, F). This accumulation seemed to be driven by KSHV infection, as KSHV-specific CD4+ T cell frequencies correlated with KSHV but not with EBV viral loads (Fig. 4G).

This aligns with the preferential homing of GFP+ B cells to the peritoneal cavity over the spleen (Fig. S7C), the formation of KSHV-associated PEL in body cavities such as the peritoneum and the identification of B cells as the main host cells for KSHV infection in double-infected humanized mice (Fig. S7D)[31–33]. GFP+ cells from EBV and KSHV-infected humanized mice were shown to contain significantly more KSHV DNA, making GFP a useful marker for KSHV-infected cells in vivo[33]. This specificity for KSHV can be attributed to GFP expression from rKSHV.219 under the constitutively active EF-1α promoter, as opposed to recombinant GFP expression from EBV p2089 under the CMV promoter, which is prone to epigenetic silencing[54–56]. Such a correlation between specific T cells and KSHV viral loads or an accumulation at KSHV-infected sites was not observed for transgenic CD8+ T cells (Figs. 4F and S7E). However, a substantial expansion in absolute numbers of CD4+ transgenic T cells appeared unlikely, as maximally 25% of the initially injected total cell count was recovered in the peritoneal lavage and up to 5% in the spleen, which is far below the approximately 15% of total body lymphocytes residing in the spleen[57,58]. A protective function, like a reduction in the frequency of KSHV-infected animals, was not observed (Fig. 4H). However, the animal numbers were not designed for nor sufficient to detect such an effect. In summary, we observed no expansion of KSHV-specific T cells. Instead, specific CD4+ T cells redistributed to sites of KSHV persistence, particularly the peritoneal cavity.

## LANA-specific CD4+ T cells acquire an activated and early differentiated effector memory phenotype in vivo

KSHV seroconversion in humans does not result in noticeable changes in T cell numbers or bulk phenotype, and immune alterations in EBV & KSHV co-infected humanized mice are predominantly driven by EBV. However, the transgenic TCR enabled us to specifically compare T cell differentiation and activation between KSHV-specific T cells and bulk T cells during primary KSHV infection in EBV co-infected humanized mice. Our analysis focused on the peritoneum, the preferential homing site for both KSHV-infected B cells and KSHV-specific CD4+ T cells. Uniform Manifold Approximation and Projection (UMAP) visualization and FlowSOM clustering were calculated based on the expression of T cell markers measured via flow cytometry and included data from seven animals that received Cl33 TCR-transduced T cells (Figs. 5A and S8A). mCD19 and mCB, which identify transgenic cells, were excluded from clustering and visualization analyses. This approach enabled the annotation of all T cells regardless of their transgene expression and allowed for the phenotypic projection of the TCR transgenic T cell population on this T cell heterogeneity. Clusters were coarsely annotated based on lineage markers and differentiation markers such as CD45RA, CCR7, CD27 and CD28 and further subdivided based on the expression of activation markers like PD1 or HLA-DR (Figs. 5B, and S8B). KSHV-specific CD4+ T cells were identified via mC19 expression, and the majority was found to be enriched in an early differentiated effector memory T cell cluster EM1, characterized by co-expression of CD27 and CD28 and the lack of CD45RA and CCR7 (Fig. 5C, D and S8C). In contrast to the untransduced bulk CD4+ T cells, further differentiated effectors lacking CD27 were rare and Tregs were absent. The accumulation of the specific cells in the CD27+ compartment at the expense of the CD27- compartment was also true on the level of individual mice, with a significantly increased frequency of CD27+ Tem and decreased frequency of CD27- Tem in the Cl33 transduced cells (Fig. 5E). Two mice had to be excluded, because less than ten KSHV-specific cells were detected. Notably, these were the two mice without detectable KSHV viral loads throughout the experiment. Similar trends were observed by manual gating on CD4+CD45RA-CCR7- effector memory T cells (Figs. S7B and S8E). Analysis of the refined expression profile of these EM1 cells showed a heterogeneous composition between individual subjects: cells either acquired a cytotoxic profile with upregulation of Granzyme B and Perforin or were characterized

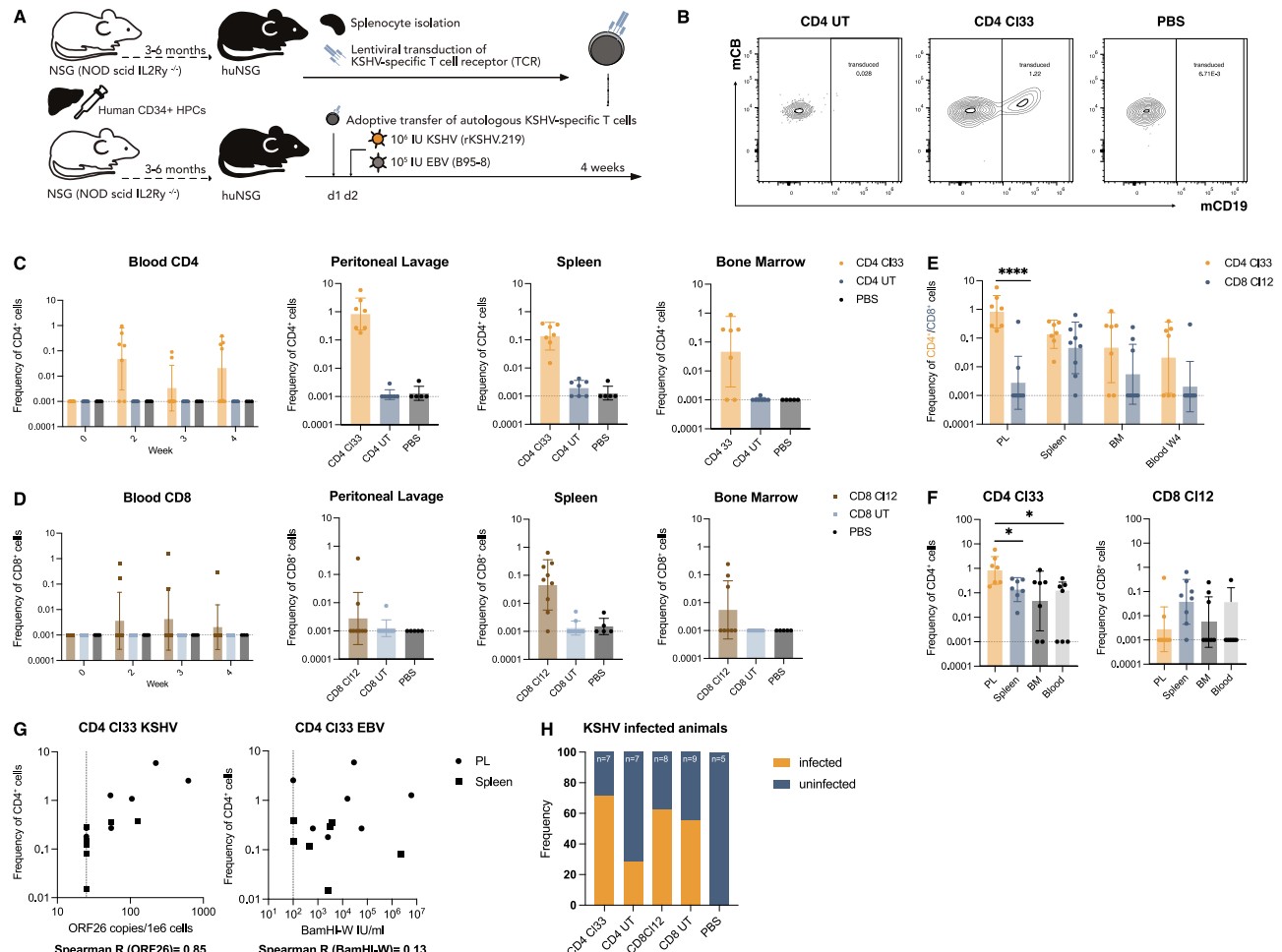

**Fig. 4 | LANA-specific CD4⁺ T cells accumulate in the peritoneal cavity of KSHV-infected humanized mice. A** Experimental outline: TCR-transduced T cells were generated and adoptively transferred into HPC donor-matched humanized mice one day before EBV&KSHV-infection. Mice were monitored over 4 weeks. **B** Flow cytometry gating strategy identifying TCR-transgenic T cells in EBV&KSHV-infected mice receiving Cl33 TCR-transduced CD4⁺ T cells (CD4 Cl33). These cells were absent in PBS controls and mice receiving untransduced T cells (CD4 UT) **C, D** Frequencies of Cl33 (CD4⁺, C) or Cl12 (CD8⁺, D) TCR-transgenic T cells in blood, spleen, bone marrow, and peritoneal lavage. Geometric mean ± geometric SD of 2 independent experiments. Groups: EBV&KSHV-infected animals receiving transgenic T cells (CD4 Cl33 (n = 7), CD8 Cl12 (blood weeks 1–2 and spleen n = 9, rest n = 8)), untransduced CD4⁺ T cells (CD4 UT; blood week 3 n = 6, week 2 n = 5, rest n = 7), untransduced CD4⁻ T cells (CD8 UT; blood week 4 n = 8, rest n = 9) or PBS mock-infected control animals (PBS; blood week 3 n = 4, week 4 n = 3, rest n = 5). **E** Comparison of transgenic T cell enrichment between CD4 and CD8 transferred

animals in peritoneal lavage (PL), spleen, bone marrow (BM) and blood at the endpoint (Blood W4). Frequencies of mCD19⁺ cells shown as geometric mean ± geometric SD of 2 independent experiments (n = 7 CD4 Cl33, n = 9 CD8 Cl12). Ordinary two-way ANOVA on log₁₀-transformed frequencies with Šidák's multiple comparisons test. **F** Organ-specific enrichment of mCD19⁺ T cells in CD4 Cl33 and CD8 Cl12 recipients. Geometric mean ± geometric SD of two independent experiments (n = 7 CD4 Cl33, n = 9 CD8 Cl12). Repeated-measures one-way ANOVA with Geisser-Greenhouse correction on log₁₀-transformed data followed by Holm-Šidák's test for matched comparisons. **G** Nonparametric Spearman correlation between viral loads (KSHV ORF26 or EBV BamHI-W) and Cl33 transgenic CD4⁺ T cell frequency. Viral loads below detection were set to half the detection limit (dashed lines). p (two-tailed)=0.0003 (ORF26), 0.66 (BamHI-W). **H** Frequency of KSHV-infected animals per group, defined by ORF26 detection in any organ at any given timepoint or LANA⁺ cells in FFPE sections. Chi-square test. Exact significant p ≥ 0.0001 (top to bottom): (4F) p = 0.0467, p = 0.0468.

by high expression of ICOS (Figs. 5F, G, and S8B, D). Some of these ICOS^high cells might still be on the way to effector memory cells, as CCR7 was slightly higher compared to other clusters (Fig. S8B). Common to ICOS^high and cytotoxic cells though was a high PD1 but rather weak HLA-DR expression. Overall, transgenic cells showed a higher mean expression of Ki-67 than the untransduced bulk CD4⁺ T cells, although not statistically significant (Figs. 5G and S8E). In summary, KSHV-specific CD4⁺ T cells accumulating at the site of infection remain early differentiated and maintain CD27 expression, while they might acquire a cytotoxic profile in individual subjects.

## Discussion

This study presents publicly available sequences of CD4⁺ and CD8⁺ KSHV-specific TCRs and demonstrates that KSHV-specific TCR-transduced T cells exhibit functional avidities and effector functions

comparable to the T cell clones from which the TCRs were isolated, including poor recognition of KSHV-infected and natively LANA-expressing B cells. Nonetheless, we could show a KSHV-dependent accumulation of CD4⁺ LANA-specific, activated and early differentiated effector memory T cells at infection sites during KSHV infection of B cells in vivo. Hence, this study highlights the localization and differentiation of latent-antigen-specific CD4⁺ T cells during primary KSHV-infection of B cells in humanized mice, and points to more efficient MHC class II presentation of viral antigens to CD4⁺ T cells in vivo than by infected B cells in vitro.

B cells, as professional APCs, express both MHC classes and thus directly present antigens to T cells. Similarly, LCLs, EBV-transformed B cells, are able to directly present viral antigens on their MHC molecules. For EBV, proteome-wide screens revealed an infection phase-dependent hierarchy of immunodominant CD8⁺ T cell targets, with

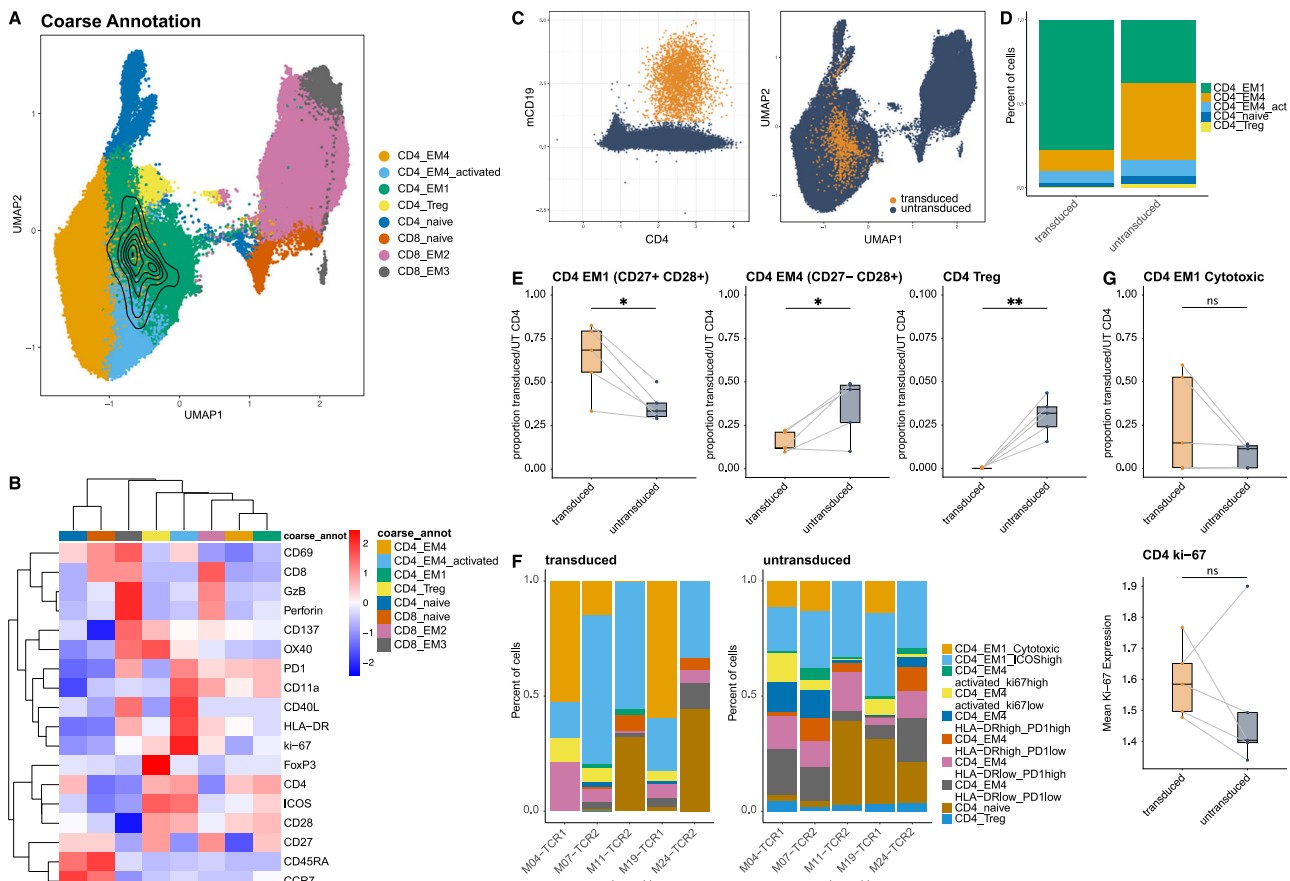

**Fig. 5 | LANA-specific CD4+ T cells acquire an early differentiated effector memory phenotype in vivo. A** UMAP visualization and FlowSOM clustering based on markers expressed on CD3+ T cells from seven animals injected with Cl33 TCR-transduced T cells from two independent experiments. mCD19 and mCB were excluded from the UMAP and clustering analysis, but mCD19+ T cells are highlighted by an overlaid contour plot on the UMAP. **B** z-score normalized average expression of T cell markers per coarse annotated cluster. **C** Scatterplot of CD4 vs mCD19 normalized expression highlighting the defined transgenic Cl33 TCR-transduced T cells in orange, characterized by mCD19 expression >1 and CD4 expression levels>1.25. UMAP shows Cl33 TCR-transduced T cells in orange. **D** Stacked bar plot of frequencies of coarse cluster annotations per cell within TCR-transduced T cells and untransduced CD4+ T cells. **E** Frequency of cells within the EM1, EM4, and Treg cluster within transduced or untransduced CD4+ cells. Graphs

show boxplots and individual values of five animals from two independent experiments. Two animals with <10 transduced cells were excluded from the analysis. Paired two-tailed t-test, two-sided p-value. **F** Stacked bar plot of frequencies of fine cluster annotations per cell within TCR-transduced T cells and untransduced CD4+ T cells per individual mouse (donor_id). **G** Frequency of cells within the EM1 cytotoxic cluster within transduced or untransduced CD4+ cells (top panel). Mean Ki-67 normalized expression within transduced or untransduced CD4+ cells (bottom panel). Graphs show boxplots and individual values of five animals from two independent experiments. Two animals with <10 transduced cells were excluded from the analysis. Paired t-test, two-sided *p*-value. (**A–G**) *\*p* < 0.05, *\*\*p* < 0.01, *\*\*\*p* < 0.001, *\*\*\*\*p* < 0.0001. Exact significant *p* ≥ 0.0001 from left to right (5E) *p* = 0.014, *p* = 0.032, *p* = 0.0031.

immediate early being more readily recognized than early or late lytic antigens[6,59–61]. This parallels with the expression efficiency of these antigens and the increased viral restriction of antigen presentation with progression through the lytic cycle and suggests that EBV-directed responses are driven by direct antigen-presentation of B cells to T cells. In contrast, CD4+ EBV-specific responses primarily target latent antigens[62]. For KSHV, no such hierarchy exists, and the most frequently recognized antigens are K8.1 and LANA[15,16]. This raises the question of whether KSHV infected B cells are capable of presenting viral antigens to T cells at all. While PEL cells are highly differentiated and known to be poorly recognized by T cells irrespective of the antigen presented, we found that freshly established EBV and KSHV co-infected B cells also failed to elicit a LANA-specific T cell response[44,45]. Although higher IFNγ expression was observed, it was not TCR-dependent. A similar observation was made in KSHV-only infected B cell models and linked to immune escape mechanisms such as LANA's acidic repeat region preventing antigen processing prior to peptide translocation to the endoplasmic reticulum for efficient presentation on MHC class I molecules, or LANA inhibiting MHC class II transactivator (CIITA) for efficient expression of MHC class II molecules[63–67].

These immune evasion mechanisms by LANA might apply to EBV and KSHV co-infected B cells as well, but might still allow recognition of KS and MCD tumor cells by LANA specific T cells.

However, despite poor in vitro recognition of infected B cells, LANA-specific CD4+ T cells accumulated at the site of infection in vivo. KSHV-directed CD4+ T cells, injected intravenously, specifically localized to the peritoneum, notably one of the main sites for PEL and therefore, a site of persistence for KSHV-infected B cells[27]. This pattern of recruitment suggests that TCR-independent bystander activation, as observed by us in vitro with freshly KSHV infected B cells, is unlikely the sole mechanism to attract LANA-specific CD4+ T cells in vivo. It further implies either enhanced antigen-presentation capacity by infected B cells in vivo or the involvement of other APCs such as dendritic cells (DC) in activating LANA-specific T cells, as has previously been reported for EBV specific T cell responses[68]. Conventional and plasmacytoid DCs (pDCs) are reconstituted in humanized NSG and NRG mice to similar frequencies as in humans[69–72]. They were found to mature upon TLR agonist stimulation and to be capable of priming CD4+ T cell responses[73]. Furthermore, a drop in numbers of pDCs in blood upon

primary EBV infection suggests their redistribution into tissues[74]. The lack in hierarchy of KSHV antigens recognized by T cells as observed for EBV might support predominant antigen-presentation by APCs other than B cells. Going forward, it might be interesting to analyze the transcriptome of infected cells and the broader immune landscape, beyond CD4+ T cells, in the peritoneum in comparison to other organs in further detail to identify additional antigen presentation mechanisms that might contribute to more efficient MHC class II restricted antigen presentation in vivo than by KSHV infected B cells in vitro.

Surprisingly, T cells were detected only transiently and at low frequencies in the blood, which presents a great contrast to the peritoneum. While EBV-specific CD4+ T cells also do not expand in numbers, their frequencies can increase to up to 1% of the total CD4+ T cell in blood[62,75]. Blood is still the most analyzed compartment in patients as it is easily accessible, but it does not appear to be indicative in KSHV-infection. Our model provided an opportunity to examine specific infection sites and revealed LANA-specific CD4+ T cells within the early differentiated effector compartment in the peritoneum. Higher resolution analysis indicated that some of these cells might still be transitioning from central to effector memory phenotypes, while others had acquired cytotoxic characteristics. These results align with two studies, although lacking information about antigen-specificity, reporting enrichment of CD4+ effector memory T cells in the bronchoalveolar lavage of pulmonary KS patients and reporting a transcriptomic profile of central memory T cells including a prominent cytotoxic population of putative KSHV-specific CD4+ T cells in PBMCs of KS patients[76,77]. Other studies focusing on known antigen-specificities primarily investigated CD8+ T cells in the blood, yielding somewhat contradictory findings. KSHV-specific CD8+ T cells were mainly of a CD27+ Tem phenotype in asymptomatic KSHV-carriers, while specific T cells in asymptomatic transplant recipients controlling KSHV infection rather showed a TEMRA (CD45RA+CCR7-) phenotype[78–80]. Another study found T cells targeting latent antigens to be more associated with a Tem phenotype, while those targeting lytic antigens were more aligned with the TEMRA phenotype[81]. The fact that CD4+ T cells differentiate differently from their CD8+ counterparts seems rather exceptional among herpesviruses. EBV-specific CD4+ and CD8+ T cells both accumulate in the early differentiated CD45RA-CCR7+ central memory and CD27+ Tem compartment while CMV-specific CD4+ and CD8+ cells both accumulate in the CD27-CD28- Tem and mainly TEMRA compartment[5,61,82–85]. Regarding CD4+ cytotoxic T cells (CTLs), these are predominantly associated with late differentiated T cell states found in persistent infections like CMV, Hepatitis or HIV[86–90]. Interestingly, we identified high frequencies of cytotoxic early Tem cells in individual mice. This observation may be plausible, given that primary EBV infection also induces early differentiated memory CD4+ T cells and CD4+ CTLs that are transcriptionally distinct from the more commonly described CTLs associated with persistent infections[91].

We further hypothesized that, akin to EBV infection and based on previously observed increased CD8+ T cell expansions during EBV and KSHV co-infection in humanized mice, LANA-specific CD8+ T cells would expand during primary KSHV infection[31]. LANA-specific CD8+ T cells are regularly identified in patients and their concurrent appearance along with clinical improvement of KS patients following reduced immunosuppressive treatment has suggested that high levels of LANA-specific CD8+ T cells responses might confer protection against KS oncogenesis[92–95]. However, they were not found to be recruited to inflamed KS tissue, raising questions about their protective role[80]. Furthermore, these cells do not appear to prevent MCD development, as similar frequencies of LANA-specific CD8+ T cells have been reported in both healthy KSHV carriers and MCD patients[78]. In humanized mice, where only B cell infection but not epithelial or endothelial cell infection can be modeled, we did not detect any expansion of CD8+ LANA-specific T cells. Although our study focused

on a single TCR, our finding suggests that LANA-specific CD8+ T cells have difficulties to be activated by LANA expressed in B cells, either directly or via cross-presentation. By contrast, we previously observed lytic K6-specific CD8+ T cells to directly recognize KSHV-infected B cells[31]. This highlights the potential interest in further investigating CD8+ T cell control of KSHV infection in B cells, particularly targeting lytic rather than latent antigens such as LANA.

Consistent with LANA being among the two most frequently recognized KSHV antigens for class switched antibody responses and possibly mainly CD4+ T cell recognition[15,21], we observed that CD4+ but not CD8+ T cells recognizing LANA follow KSHV infected B cells to the peritoneal cavity and get activated as well as differentiated into early effector memory T cells at this anatomical site. Although our study did not provide definitive evidence for a protective role of these LANA-specific CD4+ T cells, their localization, activation and differentiation suggest contribution to antiviral immunity. Notably, while adoptive transfer of EBV-specific T cells in humanized mouse models does not clearly result in measurable viral control after 5 weeks, such therapies have shown clear clinical efficacy in patients with EBV-associated diseases. This highlights that, despite the lack of definitive conclusions about the protective function of LANA-specific CD4+ T cells in our study, a clinical benefit cannot be excluded. Future experiments will reveal immune regulation or possibly direct anti-viral functions of these LANA-specific CD4+ T cells in the PEL tumor microenvironment.

## Methods
### Key resource table
See supplementary Table 1.

### EBV and KSHV production
Non-recombinant, GFP-negative EBV was purified from B95-8 cells induced with 40 ng/mL phorbol-12-myristat-13-acetat (PMA). Cells were maintained in RPMI 1640 (Gibco, Thermo Fisher) supplemented with 10% fetal bovine serum (FBS), 50 U/mL Penicillin/Streptomycin (Thermo Fisher) and 1% glutamine (Thermo Fisher) (R10) and the supernatant was harvested five days after induction. Recombinant EBV B95-8-GFP (EBV wt, p2089) was produced in HEK293 cells by inducing the lytic cycle through transfection with plasmids encoding BZLF1 and BALF4 and recombinant KSHV-GFP (rKSHV.219) was generated from latently infected BJAB cells (BrK.219) by inducing lytic gene expression using anti-IgM antibody (0.625μg/mL, Southern Biotech) and PMA (0.05μg/mL, Sigma-Aldrich)[55,96,97]. Infectious virus units (IU) of filtered and concentrated cell supernatant were determined by flow cytometric analysis of GFP-positive Raji cells (EBV) or HEK293T cells (KSHV) 48 h after infection on a FACSCanto II (BD Biosciences), as previously described[98]. Non-recombinant, GFP-negative EBV was used in the smallest volume, transforming primary CD19+ magnetic-activated cell-sorted (MACS) B cells (Milteny, 130-050-301).

### Cells, media, peptides
PBMCs were isolated from healthy donors by density gradient centrifugation (Ficoll®-Paque Premium, GE17-5442-03, Merck) and primary cells were isolated via positive selection using Miltenyi microbeads according to the manufacturer's instructions. LCLs were generated from MACS-isolated CD34-CD19+ human fetal liver- or CD19+ PBMC-derived B cells. EBV and EBV plus KSHV-infected B cells were established by spinfecting B cells with a multiplicity of infection (MOI) of 0.5 of KSHV (800 g, 4 °C, 20 min) and the smallest volume transforming B cells of non-recombinant GFP-negative EBV (800 g, 4 °C, 60 min)[31]. KSHV-infected B cells were selected by gradually increasing the Puromycin concentration starting from 0.05 μg/mL until >90% of cells expressed GFP, and the cells could be maintained with 3.2 μg/mL Puromycin. HLA-typing of the LCLs was performed at the University of Messina or the University Hospital Zurich by sequence-specific oligonucleotide PCR analysis. LCLs, T2B35 cells (kindly provided by Dr. Rajiv

Khanna, Brisbane, Australia), MC116 and VG-1 cells (both kindly provided by Dr. Alexander Hahn, Göttingen, Germany) and Raji cells were maintained in R10. Jurkat-Lucia™ NFAT reporter cells (InvivoGen) were maintained in IMDM (Gibco, Thermo Fisher) supplemented with 10% FBS, 50 U/mL Penicillin/Streptomycin and 1% glutamine (I10). Every other passage, 100 µg/mL Zeocin was added. HEK293T cells were maintained in DMEM (Gibco, Thermo Fisher) supplemented with 10% FBS, 50 U/mL Penicillin/Streptomycin and 1% glutamine (D10). iSLK.219 cells were cultured in presence of hygromycin (1 mg/mL), puromycin (1 µg/mL) and Geneticin (250 µg/mL) and lytically induced with doxycycline (1 µg/mL) and sodium butyrate (1 mM)[99]. Peptides were purchased from peptides & elephants and dissolved in $H_2O$.

## Primary T cell clones

Primary KSHV-specific T cell populations previously described in ref. 24,25 were kindly provided by Dr. Jianmin Zuo, Birmingham, UK. Thawed cells were expanded on irradiated feeder cells (100'000 PBMCs and 10'000 cognate peptide-pulsed LCLs per 96-well plate well) in the presence of 150 U/mL hrIL-2 in RPMI supplemented with 10% human serum (Corning), 5% Penicillin/Streptomycin (Thermo Fisher) and 5% L-Glutamine (Thermo Fisher) as previously described[31]. RNA of samples harvested on day 0 and day 14 after thawing the cells was isolated using the RNeasy Mini kit (QIAGEN) according to the manufacturer's instructions and used for TCR sequencing. Specificity of the T cell population for the cognate antigen described in ref. 24 was assessed on day 14. T cells were co-cultured with R10 only, PMA (25 ng/mL) plus Ionomycin (325 nM), cognate peptide-pulsed and unpulsed autologous LCLs (Cl33, Cl156, Cl114, Cl55, Cl87) or T2B35 cells (Cl12) at an effector to target ratio (E:T) of 1:1. Cells were cultured for 1 h in R10 containing only PerCP-Cy5.5-labeled mouse anti-human CD107a antibody (Clone H4A3, Biolegend) and for further 5 h containing Brefeldin A (Sigma-Aldrich) in a 1:2000 dilution in addition. To determine the TCR Vß of the degranulating cells, a multitude of co-cultures with cognate peptide-pulsed LCLs was set-up, each stained after 6 h with a unique combination of the following TCR Vß antibodies (Beckman Coulter): Vß1, Vß2, Vß4, Vß5.3, Vß7.1, Vß9, Vß12, Vß13.1, Vß14, Vß18, Vß20, Vß23 (all PE-labeled), Vß3, Vß5.1 Vß5.2 Vß8, Vß11, Vß13.6, Vß16, Vß17, Vß21.3, Vß22 (all FITC-labeled). All samples were additionally stained for hCD3 (BV785, OKT3, Biolegend), hCD8 (PE-Cy, RPA-T8, Biolegend), hCD19 (APC, HIB19, BD), hCD4 (APC-Cy7, RPA-T4, Biolegend), Zombie Aqua (Biolegend) and acquired on a Cytek Aurora 5 L spectral flow cytometer.

## TCR sequencing

The TCRα and TCRß chains from all samples were sequenced using an open-source TCR-library preparation and computation analysis pipeline described by the Chain lab at UCL London[100–102]. For sequencing, the prepared libraries were processed on an Illumina MiSeq platform with a paired end reads configuration of 201-8-8-101. The Decombinator software v4.2 available on Github (https://github.com/innate2adaptive/Decombinator) was used to analyze the sequencing results. TCRα and TCRß sequences of clone 114 were obtained through single cell RNA sequencing (scRNA-Seq). scRNA-Seq of up to 10'000 freshly thawed T cells was performed after processing the cells with the 10X Genomics Chromium Next GEM Single Cell 5' Reagent Kits v2 (Dual Index) on a Illumina Novaseq 6000. Sequencing was performed by the Functional Genomics Center Zurich. Loupe VDJ Browser (Cell Ranger, 10X genomics) was used to evaluate the TCR sequences of clone 114.

## Generation of lentiviral TCR constructs

The TCR variable α and ß chain of clone 114 were introduced into a synthetic TCR construct on the retroviral pMP71 vector previously described by Thomas et al.[40]. It was engineered to contain codon-optimized mouse constant domains with additional disulfide bonds.

TCR α and ß chains are separated by a viral P2A sequence, and the TCR ß chain is separated from a truncated mouse CD19 by a viral T2A sequence. TCR inserts flanked by Gibson homology regions and 5' Not1, BSRG1 and Agel restriction sites were synthetized by Integrated DNA Technologies (IDT) and introduced into the Not1 (5') and PmlI (3') digested pMP71vector using the Gibson Assembly® Cloning Kit (NEB) according to the manufacturer's instructions. To transfer the construct from the pMP71 into a pLV vector (kindly provided by Gregor Hutter), the construct was PCR-amplified with primers containing pLV homology regions and introduced into the BSRG1-digested pLV backbone using the Gibson assembly Cloning Kit (NEB). All other newly synthetised TCR constructs containing Gibson homology regions were introduced into the Agel and Pml-digested pLV vector using the Gibson Assembly® Cloning Kit (NEB). TCR expression is driven under the EFS promoter.

## Lentiviral production

For the lentivirus production, fresh HEK293T cells were transfected with the TCR transgene containing pLV plasmid, the packaging plasmid p8.91 (Addgene plasmid #187441 from Simon Davis, http://n2t.net/addgene:187441; RRID:Addgene_187441) and the envelope plasmid VSV.G (Addgene plasmid #14888 from Tannishtha Reya, http://n2t.net/addgene:14888; RRID:Addgene_14888) at a ratio of 3:2:1 using PEI MAX® (Polysciences). After 24 h, the media was changed to OptiMEM + 50 U/mL Penicillin/Streptomycin (Gibco, Thermo Fisher) and after a total of 48 h, supernatant was harvested, concentrated with PEG-it (System Biosciences) according to the manufacturer's instructions and frozen at −80 °C until further usage.

## TCR transduction

0.5 million T cells were transduced in 500 µL R10 in a 24-well plate by spinfection (1.5 h at 32 °C) in the presence of 8µg/mL Polybrene and 10 µL of concentrated lentivirus if not otherwise stated. 48 h prior to transduction, primary T cells were activated with 8.75 µL CD3/CD28 Dynabeads (Thermofisher Scientific) per million cells in cytokine media (R10 containing 132 U/mL rhIL-2 (Peprotech), 155 U/mL rhIL-7 (Miltenyi Biotec) and 290 U/mL rhIL-15 (Biolegend)). Fresh cytokine media was added directly after the transduction and the day after to the primary T cells, and was then replaced every other day until cells were used in functional assays 8–12 days later. Transduction efficiency and TCR expression rate were assessed earliest 48 h after transduction. Cells were stained for anti-mouse CD19 (mCD19, PE, 1D3/CD19, Biolegend), anti-mouse TCR ß chain (mCB, BV510, H57-597, Biolegend) and Zombie NIR (Biolegend), and in case of primary T cells also for hCD3 (BV785, OKT3, Biolegend), hCD8 (PerCP, SK1, Biolegend), hCD4 (PB, RPA-T4, Biolegend) and hCD19 (APC, HIB19, BD). Samples were acquired on a LSRFortessa (BD Biosciences).

## Jurkat Lucia NFAT reporter assays

Transduced Jurkat Lucia NFAT reporter cells were sorted for live, single, mouse CD19+, and mouse TCR ß+ cells on a BD Aria III 5 L or on a BD S6 5 L. For all co-culture assays, 100'000 Jurkat Lucia NFAT cells were incubated with 100'000 HLA-matched or HLA-mismatched LCLs in 200 µL I10 at 37 °C in a humified incubator. LCLs were pulsed by default with 1 µM cognate peptide for at least 30 min at 37 °C if not otherwise stated. After 24 h, 20 µL of supernatant was mixed with 50 µL 1x QUANTI-Luc™ 4 Lucia/Guassia (invivoGen) and Luminescence was measured.

## Immunoblotting

1 million cells were resuspended in 500 µL Laemmli buffer (20% PAGE 5X [20% sodium dodecylsulfate (SDS), 0.3125 M Tris-HCl in deionized water, adjusted to pH 6.8-7], 20% BBQ [50% glycerol, 20% PAGE 5X, 1% bromophenol blue in deionized water], 1.4% ß-mercaptoethanol in deionized water). Samples were boiled at 95 °C for 5 min, and proteins

were separated by SDS-polyacrylamide gel electrophoresis (PAGE) followed by a transfer to a polyvinylidene difluoride (VWR, 10600023) membrane. Membranes were stained with anti-HHV8 LANA antibody (LN53, abcam, ab4103) and anti-Tubulin (DM1A, Novus) at 4 °C overnight and incubated with peroxidase-conjugated goat anti-rat secondary antibody (Jackson ImunoResearch, 112-035-003) at 1 h at RT. WesternBright Sirius HRP kit (advansta, K-12043-D20) was used as a substrate, and the blots were imaged on a Fusion FX (Vilber Smart Imaging). Images were analyzed and quantified with ImageJ/Fiji.

### ELISA
If not otherwise stated, 10'000 T cells were incubated overnight with target cells at an E:T ratio of 1:5 at 37 °C in a humified incubator with 5% $CO_2$. T cells with medium only and T cells incubated with PMA (25 ng/mL) plus Ionomycin (325 nM) were used as negative and positive control, respectively. IFNɣ or TNF was measured in the supernatant according to the manufacturer's instructions (Human IFNɣ (HRP), Mabtech, 3420-1H; Human TNF (HRP), Mabtech, 3512-1H) and the signal was detected on an Infinite 200 PRO (Tecan). When specified, IFNɣ concentrations were normalized to those measured in untransduced T cells from the same donor and experimental condition and reported as fold over untransduced (UT) controls.

### Degranulation assay and intracellular cytokine staining
Degranulation assays with TCR-transduced T cells were performed as described above for primary T cell clones. Cells were cultured for 1 h in R10 containing only FITC-labeled mouse anti-human CD107a antibody (Clone H4A3, BD) and for further 5 h containing Brefeldin A (Sigma-Aldrich) in a 1:2000 dilution in addition. All samples were additionally stained for hCD3 (BV785, OKT3, Biolegend), hCD8 (PE-Cy, RPA-T8, Biolegend), hCD19 (APC, HIB19, BD), hCD4 (APC-Cy7, RPA-T4, Biolegend), Zombie Aqua (Biolegend) and acquired on a LSRFortessa (BD Biosciences). In case of intracellular cytokine staining, cells were stained for surface hCD3 (BUV395, UCHT1, BD), hCD4 (BUV496, SK3, BD), hCD8 (BUV563, RPA-T8, BD), hCD19 (BUV661, HIB19, BD), mouse TCR ß chain (BV510, H57-597, Biolegend), Zombie NIR (Biolegend) followed by fixation and permeabilization using the Cytofix/Cytoperm Fixation/Permeabilization Kit (BD Biosciences) and intracellular staining of Granzyme B (Alexa Fluor 700, QA18A28, Biolegend), IFNɣ (APC, 4S.B3, Biolegend), TNF (PE, MAB11, Biolegend) and Perforin (PerCP-Cy5.5, δG9, BD). Samples were acquired on a Cytek Aurora 5 L spectral flow cytometer.

### Killing assay
Target cells were labelled using the red PKH membrane label kit (Sigma-Aldrich) according to the manufacturer's instruction and co-cultured with T cells in an E:T ratio of 5:1 for 18 h at 37 °C in a humidified incubator with 5% $CO_2$. Labelled target cells without the addition of T cells were used as control. Cells were stained with the Zombie NIR fixable viability kit (Biolegend) and acquired on a LSRFortessa (BD Biosciences). Specific killing was calculated as follows: (Frequency of dead cells [sample]-Frequency of dead cells[control])/(100%- Frequency of dead cells[control]). In case of surface marker assessment, cells were stained for surface hCD3 (BUV395, UCHT1, BD), hCD4 (BUV496, SK3, BD), hCD8 (BUV563, RPA-T8, BD), Zombie Aqua (Biolegend) and PD1 (BUV737, EH12.1, BD), mouse TCR ß chain (B605, H57-597, Biolegend), CD40L (BV785, 24-31, Biolegend), CD244 (FITC, C1.7, Biolegend), OX40 (Alexa Fluor 647, Ber-ACT35, Biolegend), CD137 (APC-Fire750, 4B4-1, Biolegend) or HLA-DR (APC-Cy7, L243, Biolegend). Samples were acquired on a Cytek Aurora 5 L spectral flow cytometer.

### Proliferation assay
Target cells pulsed with 1 µM of cognate peptide and unpulsed control target cells were irradiated with 60 Gy. TCR-transduced and untransduced control cells were labelled with the CellTrace™ Far Red Cell Proliferation kit (Invitrogen) according to the manufacturer's instructions. 100'000 labelled T cells and 100'000 irradiated target cells were resuspended in 200uL RPMI 1640 (Gibco, Thermo Fisher) supplemented with 10% human serum AB male (BioConcept), 50 U/mL Penicillin/Streptomycin (Thermo Fisher) and 1% glutamine (Thermo Fisher) (H10) and cultured for 1 week at 37 °C in a humidified incubator with 5% $CO_2$. Labelled T cells without target cells were kept as negative controls in the presence of 125 U/mL rhIL-2 (Peprotech). Cells were stained with hCD4 (PB, RPA-T4, Biolegend), Zombie Aqua (Biolegend) or LIVE/DEAD Fixable Blue Dead Cell Stain Kit (Invitrogen), anti-mouse TCR ß chain (mCB, BV605, H57-597, Biolegend), CD25 (BV711, M-A251, Biolegend), hCD3 (BV785, OKT3, Biolegend), hCD8 (PerCP, SK1, Biolegend), anti-mouse CD19 (PE, 1D3/CD19, Biolegend), OX40 (PE-Cy7, Ber-ACT35, Biolegend), ICOS (APC-Cy7, C298.4 A, Biolegend) and acquired on a Cytek Aurora 5 L spectral flow cytometer. Proliferating CellTrace Far Red[low] cells and transduced mCD19+ cells were normalized to the background in the untransduced control cells. For CD4+ T cells, readouts were normalized to T cells co-cultured with unpulsed EBV-only infected target cells.

### B-cell recognition assays
HLA-B*35:01, HLA-DRB1*13:01 positive MC116 cells were infected with rKSHV.219 with an MOI1 or peptide-pulsed with 1 µM of cognate peptide. Lytic cycle was induced for 48 h in HLA-B*35, HLA-DRB1*13:01 positive Brk.219 cells according to the KSHV production protocol mentioned above[55] or Brk.219 cells were peptide-pulsed with 1 µM of cognate peptide. Untreated MC116 and Brk.219 cells as well as BJAB cells were kept as controls. 10'000 primary TCR-transduced T cells of four different donors were co-cultured overnight with MC116, Brk.219 or BJAB cells at an E:T ratio of 1:3 at 37 °C in a humidified incubator with 5% $CO_2$. Primary B cells were isolated from healthy donors with CD19 Miltenyi microbeads and infected with EBV B95-8 or EBV B95-8 plus rKSHV.219 as previously described for the LCL generation. 10'000 autologous TCR-transduced T cells from the same B*35:01 or DRB1*13 positive donors were co-cultured with primary B cells on day 1, 5 and 14 post-infection with an E:T ratio of 1:1. Alternatively, T cells were incubated only with the supernatant of B cells from day 5–10 post-infection. In both assays, untransduced T cells as well as T cells with medium only and T cells incubated with PMA (25 ng/mL) plus Ionomycin (325 nM) were used as negative and positive controls, respectively. After overnight incubation, IFNɣ was measured in the supernatant by ELISA as described above.

### Humanized mouse generation
NOD.Cg-$Prkdc^{scid}$ $Il2rg^{tm1Wjl}$/SzJ (Strain #005557, NSG) animals obtained from the Jackson Laboratory were bred at the Institute of Experimental Immunology, University of Zurich. Mice were humanized by injecting one to five days old, sublethally irradiated (1 Gy) pups intrahepatically with 1–3×10$^5$ human fetal liver (HFL) derived (Advanced Bioscience Resources, Cercle Allocation Services, Inc.) CD34+ hematopoietic progenitor cells (HPCs). HPCs were isolated with the CD34 MicroBead kit (Milteny) and HLA-typed[10]. The frequency of CD34+ isolated HPCs was assessed before injection by flow cytometry staining for CD34 (APC, 581, Invitrogen), CD38 (PE, HIT2, Biolegend) and Zombie Aqua (Biolegend). Three months after the HPC injection, the human immune system reconstitution was assessed in the peripheral blood by flow cytometry staining for hCD45 (PB, HI30, Biolegend), hCD3 (PE, UCHT1, Biolegend), hCD19 (PE-Cy7, HIB19, Biolegend), hCD4 (APC-Cy7, RPA-T4, Biolegend), hCD8 (PerCP, SK1, Biolegend), hNKp46 (APC, 9-E2, BD), hHLA-DR (FITC, L243, Biolegend). Each experiment consisted of animals reconstituted with HPCs from the same HFL donor, and animals were allocated by stratified randomization to the experimental groups according to their human immune reconstitution assessment

and their sex. The experiment was not designed to find a sex effect. Animals with less than 10% hCD45+ cells of all leukocytes were excluded from the study.

## Virus infection and adoptive T cell transfer

Three-month-old humanized donor mice were euthanized, and splenocytes were isolated as described below. CD4+ T cells were isolated via positive selection using CD4 Miltenyi microbeads according to the manufacturer's protocol. CD4 positive and negative cells were activated with 8.75 µL CD3/CD28 Dynabeads and transduced with the transgenic TCR as outlined above. CD4+ T cells were transduced with Cl33 TCR (CD4+ T cell-derived) and CD4- cells with Cl12 TCR (CD8+ T cell-derived). Cells were resuspended in fresh cytokine media directly after the transduction as well as on days 1 and 2 post-transduction. Transduction efficiency and TCR expression were assessed by flow cytometry on day 2 as outlined above. On day 3, 150'000-200'000 transgenic TCR+CD3+ T cells were injected intravenously into HPC donor-mate animals. Similar amounts of untransduced cells were transferred into control HPC donor-mate animals. The following day, mice were intraperitoneally injected with $10^5$ IU EBV B95-8 (p2089) and $10^6$ IU rKSHV.219. PBS control animals received PBS intravenously on day 3 and intraperitoneally on day 4. Animals were euthanized by $CO_2$ inhalation 4 weeks post-infection or if predetermined humane endpoints were met.

## Cell isolation of animal tissue

Splenocytes from mashed spleens were isolated by density gradient centrifugation at $1100 \times g$ for 25 min at room temperature using Ficoll-Paque (GE Healthcare). Bone marrow cells were isolated from tibia and femur by centrifugation of cut-open bones, giving access to the bone marrow. The bone marrow pellet was collected in 1.5 mL Eppendorf tubes and strained through a 70 µm filter. Blood for PBMC isolation was gained from heart puncture or weekly tail vein nicking. Blood was collected in EDTA tubes (BD Microtainer K2-EDTA tubes), and 55 µL whole blood was aliquoted for DNA isolation. Erythrocyte lysis was performed using an in-house ACK lysis buffer. Cells from the peritoneal cavity were collected by injecting 5 ml PBS containing 3% FBS, followed by aspiration of the lavage fluid. All counts of cells isolated from tissues, as well as whole blood cell counts, were assessed using a DxH500 Hematology Analyzer (Beckman Coulter) and aliquoted for subsequent analysis. For flow cytometry, $5 \times 10^5$–$5 \times 10^6$ cells were used for staining, while $0.5 \times 10^6$–$1 \times 10^6$ cells were used for DNA isolation.

## Quantification of EBV and KSHV DNA

DNA was extracted from whole blood using the NucliSENS® easyMag® (bioMérieux), while DNA from splenocytes, bone marrow, liver and cells derived from peritoneal lavage was extracted using the DNeasy Blood & Tissue Kit (QIAGEN), following the manufacturer's instructions. Viral DNA in blood and organs was detected using quantitative real-time PCR (qRT-PCR) targeting BamHI-W fragments of EBV and ORF26 of KSHV, as previously described[33]. qPCR was done with the TaqMan® Universal PCR Master Mix (Applied Biosystems) using the following primers and probe: EBV BamHI-W (fw) 5'-CTT CTC AGT CCA GCG CGT TT-3'; BamHI-W (rev) 5'-CAG TGG TCC CCC TCC CTA GA-3'; BamHI-W (probe) 5'- (FAM) CGT AAG CCA GAC AGC AGC CAA TTG TCA G(TAMRA) -3'; KSHV ORF26 (fw) 5'-GCTCGAATCCAACGGATTTG-3'; KSHV ORF26 (rev) 5'-AATAGCGTGCCCCAGTTGC-3'; KSHV ORF26 (probe) 5'-(FAM)-TTCCCCATGGTCGTGCCTC-(BHQ-1)-3'. Samples were analyzed in triplicate on the CFX384 Touch Real-Time PCR Detection System with the following protocol: 2 min at 50 °C, 10 min at 95 °C, 50 cycles of 95 °C for 15 s, 60 °C for 1 min. Animals were considered KSHV-uninfected if no KSHV DNA (ORF26) was detected in spleen, blood, bone marrow, or peritoneal lavage, and no LANA-positive cells were identified in FFPE tissue sections.

## Flow cytometry analysis

All humanized-mouse derived samples were stained with the following antibodies: CD45 (BUV395, HI30, BD), CD154 (BUV496, 24-31, BD), CD4 (BUV563, SK3, BD), HLA-DR (BUV661, G46.6, BD), PD-1 (BUV737, EH12.1, BD), ICOS (BUV805, DX29, BD), CD27 (BV421, O323, Biolegend), CD69 (Pacific Blue, FN50, Biolegend), Zombie Aqua (Biolegend), Ki-67 (BV605, ki-67, Biolegend), anti-mouse TCR beta (BV650, H57-597, Biolegend), CD3 (BV711, OKT3, Biolegend), CD45RA (BV785, HI100, Biolegend), FoxP3 (FITC, 259D, Biolegend), CD8 (Spark Blue 550, SK1, Biolegend), CD11a (PerCP, TS2/4, Biolegend), Perforin (PerCP-Cy5.5, δG9, BD), CD19 (PE, HIB19, Biolegend), CCR7 (PE-Dazzle5994, G04H7, Biolegend), CD28 (PE-Cy5, CD28.2, BD), anti-mouse CD19 (PE-Cy7, 1D3/CD19, Biolegend), IFNγ (APC, 4S.B3, Biolegend), OX40 (Alexa Fluor 647, Ber-ACT35, Biolegend), Granzyme B (Alexa Fluor 700, GB11, BD), CD137 (APC-Fire750, 4B4-1, Biolegend). Super Bright Complete Staining Buffer (eBioscience) was included in the master mix in order to prevent polymer interactions. Single cell suspensions were stained for 30 min at 4 °C for surface antigens, followed by fixation and permeabilization with the Foxp3 staining kit (eBioscience) according to the manufacturer's instructions. Samples were acquired and unmixed on a Cytek Aurora 5 L spectral flow cytometer. SpectroFlo Software (v3.0) was used for QC, sample acquisition and spectral unmixing. Manual gating was performed with FlowJo™ Software (v10, BD) and unmixed, quality-controlled, single, live, human CD45+CD3+ cells were exported. For UMAP visualization and clustering analysis, FCS files were analyzed using the Cyclone R-based pipeline available on GitHub[103]. Channels for mCD19, mCB, IFNγ, CD45, CD19, CD3, Zombie Aqua and GFP were excluded from the clustering analysis and UMAP visualization, and the following parameters were used: data processing (arcsinh_cofactor = 6000), UMAP (n_neighbors = 15; min_dist = 0.1; spread = 0.1; learning_rate = 0.5), clustering (k = 3), FlowSOM (x-dim = 3, y-dim = 7). The data was organized into a SingleCellExperiment object[104], annotated and subsequently analyzed using dittoSeq[105]. TCR-transgenic CD4+ T cells were identified as having mCD19 expression levels >1 and CD4 expression levels>1.25.

## Immunohistochemistry

Spleen and liver sections, as well as mesenteric lymph nodes, were fixed in 4% formalin and paraffin-embedded (FFPE). Immunohistochemical staining for EBNA2 (clone PE2, Abcam) and LANA (clone LN53, Clinisciences) was performed on a Leica BOND-III automated IHC staining system. Co-stainings for LANA with hCD3 or hCD19 were performed by Sophistolab AG (Muttenz, Switzerland) using the following antibodies: LANA HHV8 (clone LN53, Clinisciences, detection with AP/FastRed), CD19 monoclonal antibody (clone 6OMP31, eBioscience, detection with HRP/DAB), and CD3 monoclonal antibody (clone SP7, Diagnostic BioSystems, detection with HRP/DAB). The slides were acquired with the PerkinElmer Vectra 3.0 automated quantitative pathology imaging system using Phenochart (v1.0) and InForm (v2.4.8) software.

## Quantification and statistical analysis

Unpaired, parametric data were analyzed by a two-tailed t-test. Paired data of functional in vitro assays were analyzed for normal distribution using the Shapiro-Wilk test, followed by a paired two-tailed t-test, taking the donor of the T cells and target cells into account. Matched data with more than one group was analyzed using repeated-measure one-way ANOVA with Geisser-Greenhouse correction, followed by Holm-Šidák's multiple comparisons test. Comparisons of continuous data with more than one factor were done using two-way ANOVA. Comparison of continuous data, assuming normal distribution, across more than two groups was conducted using an ordinary one-way ANOVA followed by Tukey's multiple comparison test or using a linear mixed model fitted by REML, accounting for the donor of the different

target cells as a random effect (R Core Team 2021, lme4 package, v1.1, Bates 2015). Post-hoc pairwise comparisons were performed using the Tukey test to assess differences between groups (R Core Team 2021, multcomp package, v1.4, Hothorn 2008). P-values were adjusted using the Holm method. $EC_{50}$ values were determined using nonlinear regression with a log(agonist) versus response model with variable slope (four parameters). Correlation was assessed by Spearman's rank test. Chi-square test was used to compare proportions. In all boxplots shown, the box spans from the first to the third quartile, and the median is shown as a horizontal line. Whiskers extend to the maximum data points within 1.5 times the interquartile range from the quartiles. All datapoints, including outliers, are overlaid as individual points. All analyses were performed using GraphPad Prism or R Statistical Software (v4.2.0; R Core Team 2021).

### Ethical statement

The use of peripheral blood samples from healthy donors with appropriate informed consent, as well as the use of human fetal liver tissue, was authorized by the cantonal ethical committee of Zurich, Switzerland (KEK-ZH-NR. 2019-00837). The animal experiments performed were approved by the veterinary office of the canton of Zurich, Switzerland, in the licenses ZH212/2020 and ZH192/2023.

### Reporting summary

Further information on research design is available in the Nature Portfolio Reporting Summary linked to this article.

## Data availability

The raw TCR-sequencing data generated in this study have been deposited in the European Nucleotide Archive (ENA) under accession code PRJEB101731 for bulk TCR sequencing and under accession code PRJEB103950 for VDJ scRNAseq data. The processed TCR sequences corresponding to confirmed functional KSHV-specific TCRs are provided in the Source Data file. Processed data supporting the findings of this study, including the consensus annotations for Cl114 are provided in the source data file. All other data are available in the article and its Supplementary files or from the corresponding author upon request. Source data are provided with this paper.

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

## Acknowledgements

We would like to thank Dr Jianmin Zuo for providing us with patient derived primary T cell clones, Angelika Holler, Suzanne Byrnes, Sharyn Thomas, Martina Milighetti, Hans Stauss and Benny Chain (UCL London) for their help and assistance with the TCR sequencing and

cloning, the Functional Genomics Center Zurich (FGCZ) of University of Zurich and ETH Zurich, and in particular Timothy Sykes, Qin Zhang and Falko Noé for their support with the TCR sequencing analysis, Kristin Gehrmann for the generation of the humanized mice at the University of Zurich, Petra Hirschmann (Pathology University Hospital Basel) and Sophistolab for the immunohistochemical staining of the tissue sections, the cytometry facility (University of Zurich) for providing guidance with the FACS sorting, Anne Müller, Fabienne Läderach, Felix Baur, Kristin Gehrmann, Lucas Romann, Saskia von Boxberg and Svenja Kösegi (Institute of Experimental Immunology, University of Zurich) for helping processing the animals at study endpoints and Guido Ferlazzo (Universita' degli Studi di Messina) for the HLA-Genotyping. This research was supported by the Swiss National Science Foundation (310030_204470/1, 310030L_197952/1 and CRSII_222718_10000065), Cancer Research Switzerland (KFS-5896-08-2023-R), the Swiss MS Society (2023-17), the Swiss State Secretariat for Education, Research and Innovation (SERI) for EU Horizon BEHIND-MS, CRPP-ImmunoCure of the University of Zurich, the Sobek Foundation, the Swiss Vaccine Research Institute, Roche, Novartis, Pfizer and Viracta to C.M., M.B. was supported by an MD-PhD fellowship from the Swiss Academy of Medical Sciences, Switzerland (323630_19938).

## Author contributions

Conceptualization: M.B., C.M.; Methodology: M.B., A.H., D.V., H.S.; Formal analysis: M.B.; Investigation: M.B., S.P., D.V., K.H., S.N., S.S., A.V., L.R.; Resources: H.S., C.M.; Data Curation: M.B.; Writing—original draft preparation: M.B.; Writing—review and editing: all authors; Visualization: M.B., S.N.; Supervision: C.M., Project administration: M.B., Funding acquisition: M.B., C.M.

## Competing interests

The authors declare no competing interests.
