## [Transparent Peer review file · Nature Communications]

LANA-Specific CD4+ Effector T Cells Accumulate at the Site of KSHV Infection in Humanized Mice

Corresponding Author: Professor Christian Münz

Version 0:

Reviewer comments:

Reviewer #1

(Remarks to the Author)

In this interesting manuscript, Böni and colleagues isolate several LANA-specific T cell receptors from KSHV LANA-specific CD4 and CD8 T cell clones, report their sequence and study their function after transfusing them into human primary T cells and into T cells from humanised mice. Their main findings are that these TCRs from CD4 and CD8 positive LANA-specific T cells only show low killing activity against pulsed antigen-presenting B cells in vitro, that CD4 T cells transduced with LANA-specific TCRs preferentially home to the peritoneal cavity (the site of primary effusion lymphoma) and differentiate towards an effector-memory 1 phenotype. Their observations reinforce the notion that KSHV, in contrast to EBV, only elicits a weak T cell response and suggest that KSHV-infected B cells are ineffective in presenting KSHV LANA antigens to CD4 and CD8 positive T cells.

The experiments reported in this manuscript represent a technological tour de force and add important new insights to our understanding of the T cell response to KSHV. I only noticed a few smaller points that the authors may want to address:

1. Line 175/176: the authors refer to "an increased frequency of Granzyme B expressing CD4+ and CD8+ T cells upon TCR activation" in figures 2F-G. Looking at figure 2F I thought that the frequency of GrzB expressing pulsed transduced CD4 cells was, if anything, lower than the frequency of GrzB-expressing pulsed untransduced CD4 cells.

2. line 262/263: the authors state that "CD4+ transgenic T cells were consistently present in all analysed organs except the blood (Fig. 4C)". Fig 4C shows a frequency of transgenic CD4+ T cells of up to about 0.1% in blood, spleen and bone marrow, and a frequent of about 1% in peritoneal lavage cells. The frequency in blood is therefore in the same range as in spleen and bone marrow. The authors may want to rephrase this sentence. Their key point, that LANA TCR transgenic CD4+ cells home to the peritoneal cavity is not affected by this point.

3. Line 282: here the authors state that "aa reduction in the frequency of KSHV-infected animals was not observed" without referencing a figure panel. Presumably they refer to the data shown in figure 4H and may want to reference this panel.

4. In Figure 3G, the authors show a positive correlation between viral load and the frequency of transgenic CD4+ cells. Does this suggest that, rather than controlling viral load, KSHV-LANA-specific T cells expand as a result of an increased viral load/a higher number of KSHV-infected B cells and do the authors want to comment on this?

Reviewer #2

(Remarks to the Author)

This study provides extensive data documenting 5 TCRs to KSHV LANA (4 CD4 and 1 CD8). Studies include function of T cells expressing these TCRs in vitro and in vivo, in response to peptide loading or KSHV infection. An interesting conclusion is that while these T cells recognize peptide loaded into LCLs, they do not recognize B cells infected with KSHV and expressing LANA. Nevertheless, they preferentially localize into sites of infection, namely the peritoneal cavity. The discussion provides some possible explanations for this puzzling finding (which is actually consistent with prior data), including that LANA itself inhibits antigen presentation and the presence of other APCs in vivo, like DCs, are necessary. Testing these ideas experimentally would have further strengthened the paper, but on its own, it provides important novel

data. The following are suggestions for revision.

1. For C112, T2B35 cells were used, which is understandable given the antigen presentation for CD8 cells, while for all other clones, LCLs were used. It would be useful to describe the T2B35 line when first mentioned. Also, in many figures and figure legends (for example Fig 1, panel A), LCLs are described also for this clone. Unless I am misunderstanding, for C112 is should say T2B35 instead of LCLs, unless the authors say something like "antigen presenting cells (APCs)" throughout.
2. On Line 298 referring to figure 5A, "mCD19 and mCB as markers for identifying the transgenic cells were not used for clustering and visualization". While this makes sense, these markers were used in Figure 5C, so it would be useful to have an explanation for why they were excluded from the FlowSOM clustering (e.g. to annotate all the T cells regardless of transgene expression).
3. Figure 1, panel B. There are no labels on the two flow cytometry plots. The legend should define that the panel on the middle is untransduced and on the right transduced, or labels should be added to this effect on the figure.
4. Does the data suggest that in contrast to EBV, T-cell therapy for KSHV related diseases is unlikely to be successful? Albeit speculative, it might be nice to provide some thoughts on this in the discussion.
5. Very minor- perhaps replace the Cesarman 2011 reference for the review in Blood 2022 (PMID: 34479367) which is more up to date.

Reviewer #3

(Remarks to the Author)

Böni et al. describes the cloning of KSHV LANA-specific TCRs and development of the TCRs transgenic mice. However, these T cells fail to recognize KSHV-infected B cells in vitro. The CD4 T cells are detected at infection sites in the humanized mice and develop into activated and early effector memory. These observations are interesting but remain preliminary. Numerous experiments can be carried out to enhance the study, which are detailed below.

Line 25: "T cells against KSHV, however, are barely detectable in infected individuals,": Most of these studies examined immunocompromised individuals, so this is not a fair statement.

Line 46 to 82: the comparison between EBV and KSHV is unfair because these studies had different types of populations: one predominantly in immunocompetent subjects while the other predominantly in immunocompromised subjects.

LANA is known to inhibit antigen presentation. Thus, the use of antigen needs to take this into consideration.

Line 190-197: PEL cells are usually derived from undifferentiated or minimally differentiated precursor B cells with minimal expression of cell surface markers. It is unclear how these LCL-derived cells could mimic the PEL cells. Many PEL cell lines have been isolated. The authors should test these lines.

Figure 3D-E: unlike peptides, antigens from these cells need to be processed and presented in a proper way in order to be recognized by the T cells. Hence, one should not make a conclusion based on this in vitro assay.

Line 254-257: The authors use KSHV-infected mice reconstituted with human CD34+ cells to test the function of the isolated T cells. However, it is unclear what cells are infected? Are they latent or lytic? Do they develop LPDs? This model does not model PEL nor KS. The authors need to better characterize the status of infection and the impact on the infected cells.

The authors should try to confirm the presence of LANA-specific T cells in the peritoneal cavity of KSHV-infected human subjects as this observation is interesting.

Version 1:

Reviewer comments:

Reviewer #1

(Remarks to the Author)

Böni and colleagues report the isolation of several LANA-specific T cell receptors from KSHV LANA-specific CD4 and CD8 T cell clones, their sequence and study their function after transfecting them into human primary T cells and into T cells from humanised mice. The main message of their manuscript is that these TCRs from CD4 and CD8 positive LANA-specific T cells only show low killing activity against KSHV-infected B cells and that CD4 T cells transduced with LANA-specific TCRs preferentially home to the peritoneal cavity (the site of primary effusion lymphoma) and differentiate towards an effector-memory phenotype. Their observations support the observation that KSHV, in contrast to EBV, only elicits a weak T cell response and suggest that KSHV-infected B cells are ineffective in presenting KSHV LANA antigens to CD4 and CD8

positive T cells. The unique contribution of this manuscript to the field of KSHV T cell immunity is that they could analyse the function of KSHV LANA-specific T cells in a small animal model of EBV/KSHV-coinfected mice. The manuscript therefore makes an important contribution to this field.

The authors have carefully addressed the three reviewers' comments, added further experimental detail to clarify a few technical points and clarified some of the unclear issues pointed out by the reviewers. I have no further suggestions.

Reviewer #2

(Remarks to the Author)

This manuscript has been revised, and prior reviewers' concerns have been addressed. I have no further comments or suggestions.

Reviewer #3

(Remarks to the Author)

While I appreciate the authors' efforts to address several of my previous comments, I remain in disagreement with their responses to others, which I believe remain critical to the study's interpretation and impact.

Although the authors cite a few studies that examined CTL responses in endemic KS, these studies are limited in scope, with small subject numbers and narrow approaches. The more appropriate and informative comparison would be to examine KSHV-infected subjects who do not develop malignancies, as this would provide a stronger framework for understanding immune control and pathogenesis.

The above concern also directly impacts the choice of control groups (EBV). Without carefully selected controls, the conclusions regarding immune responses and viral interactions remain difficult to generalize.

While up to 70% of PEL cases are co-infected with EBV, KS is uniquely associated with KSHV. This distinction must be acknowledged more clearly, as the introduction of EBV into the mouse model could complicate the interpretation of results.

In summary, although the authors have strengthened the manuscript in some areas, these major concerns remain and should be directly acknowledged and discussed to improve the clarity and validity of the study's conclusions.

Point-by-point response

We thank the three reviewers for their encouraging and constructive comments. We have now addressed all of their concerns in the revised manuscript version and outline the changes below and by underlining in the revised manuscript text.

Reviewer #1 (Remarks to the Author):

In this interesting manuscript, Böni and colleagues isolate several LANA-specific T cell receptors from KSHV LANA-specific CD4 and CD8 T cell clones, report their sequence and study their function after transfusing them into human primary T cells and into T cells from humanised mice. Their main findings are that these TCRs from CD4 and CD8 positive LANA-specific T cells only show low killing activity against pulsed antigen-presenting B cells in vitro, that CD4 T cells transduced with LANA-specific TCRs preferentially home to the peritoneal cavity (the site of primary effusion lymphoma) and differentiate towards an effector-memory 1 phenotype. Their observations reinforce the notion that KSHV, in contrast to EBV, only elicits a weak T cell response and suggest that KSHV-infected B cells are ineffective in presenting KSHV LANA antigens to CD4 and CD8 positive T cells.

The experiments reported in this manuscript represent a technological tour de force and add important new insights to our understanding of the T cell response to KSHV. I only noticed a few smaller points that the authors may want to address:

1. Line 175/176: the authors refer to "an increased frequency of Granzyme B expressing CD4+ and CD8+ T cells upon TCR activation" in figures 2F-G. Looking at figure 2F I thought that the frequency of GrzB expressing pulsed transduced CD4 cells was, if anything, lower than the frequency of GrzB-expressing pulsed untransduced CD4 cells.

Thank you for raising this point. For clarification, we have now included the mean expression levels as well as the statistical significance for the comparison between T cells co-cultured with unpulsed versus cognate peptide pulsed target cells in the revised Figures 2F and G. As shown, untransduced T cells displayed similar frequencies of all markers regardless of whether the target cells were pulsed with the cognate peptide. In contrast, TCR transduced T cells showed a significantly increased frequency of for example Granzyme B expression upon stimulation with peptide-pulsed target cells. This difference highlights the specific antigen-dependent activation of the transduced T cells that induced among other markers higher Granzyme B expression. While, therefore, Granzyme B induction compensates for loss due to degranulation, the more limiting perforin expression leads indeed to loss after cognate target cell recognition. We have observed this also previously for BMLF1 and LMP2 specific CD8+ T cell clones (Antsiferova *et al.*, *PLoS Pathog* **10**, e1004333 (2014)). We hope this additional data clarifies the results in Figure 2F-G and provides more transparency.

2. line 262/263: the authors state that "CD4+ transgenic T cells were consistently present in all analysed organs except the blood (Fig. 4C)". Fig 4C shows a frequency of transgenic CD4+ T cells of up to about 0.1% in blood, spleen and bone marrow, and a frequent of about 1% in peritoneal lavage cells. The frequency in blood is therefore in the same range as in spleen and bone marrow. The authors may want to rephrase this sentence. Their key point, that LANA TCR transgenic CD4+ cells home to the peritoneal cavity is not affected by this point.

Thank you for pointing this out. We have revised the sentence on page 7 to more accurately reflect the data: "In contrast, CD4+ transgenic T cells were detected in all analyzed organs at mean frequencies $\geq 0.1\%$, including spleen, bone marrow, blood and peritoneum, although they were absent from the bone marrow and blood in some animals (Fig. 4C). ... Unlike TCR transgenic CD8+ T cells, KSHV-specific CD4+ T cells preferentially homed to the peritoneal cavity, comprising up to 5% of total CD4+ T cells at that site (Fig. 4E-F)". This revised wording more precisely describes the observed frequencies and maintains our key point that transgenic CD4+ T cells preferentially accumulate in the peritoneal cavity.

3. Line 282: here the authors state that "aa reduction in the frequency of KSHV-infected animals was not observed" without referencing a figure panel. Presumably they refer to the data shown in figure 4H and may want to reference this panel.

Thank you for alerting us to this omission. We have corrected the text to specifically reference Figure 4H on page 8, which displays the relevant data.

4. In Figure 3G, the authors show a positive correlation between viral load and the frequency of transgenic CD4+ cells. Does this suggest that, rather than controlling viral load, KSHV-LANA-specific T cells expand as a result of an increased viral load/a higher number of KSHV-infected B cells and do the authors want to comment on this?

Thank you for this insightful question. Indeed, our data suggests that KSHV LANA-specific transgenic CD4+ T cells accumulate at sites of infection in response to increased viral load. Whether this accumulation reflects true proliferation or merely tissue-specific recruitment remains uncertain. To address this, we assessed Ki-67 expression in TCR transgenic versus untransduced T cells isolated from the peritoneal lavage (new Figures 5G

and S7E). Ki-67 expression was higher in TCR transgenic compared to untransduced T cells in all but one animal, suggesting accumulation by proliferation. This is now discussed on page 8 of the revised manuscript.

In line with this, Nalwoga et al. found that IFN γ responses to LANA (ORF73) were increased in individuals with detectable KSHV DNA in peripheral blood mononuclear cells, indicating an expansion of IFN γ producing cells in response to elevated viral loads. (Nalwoga, A. et al., *Nat Commun* **12**, 7323 (2021)). This observation aligns with findings in EBV infection, where T cell expansion can be driven by high viral loads, such as during infectious mononucleosis, yet may still exert protective effects in adoptive transfer settings in patients.

As for the protective capacity of these cells, our current data do not allow for definitive conclusions. As shown in Figure 4H, we did not observe a reduction in the frequency of infected animals following adoptive transfer of LANA-specific T cells. However, in other studies on protective T cell responses it was necessary to transfer 10^7 cells to achieve immune control of EBV transformed lymphoma cells (Mietz et al., *EBioMedicine* **106**, 105240 (2024); Palianina et al., *Sci Adv* **10**, eado2048 (2024)). Therefore, we consider the transfer of 2×10^5 TCR transduced T cells with which we obtained CD4 $^+$ T cell frequencies of 0.1% rather as indicator specificities which allow us to study the differentiation, tissue distribution and accumulation of distinct antigen-specific CD4 $^+$ and CD8 $^+$ T cell populations. In future studies we plan to compare EBV and KSHV specific T cell populations for their protective effect against double-infected B cells.

Reviewer #2 (Remarks to the Author):

This study provides extensive data documenting 5 TCRs to KSHV LANA (4 CD4 and 1 CD8). Studies include function of T cells expressing these TCRs in vitro and in vivo, in response to peptide loading or KSHV infection. An interesting conclusion is that while these T cells recognize peptide loaded into LCLs, they do not recognize B cells infected with KSHV and expressing LANA. Nevertheless, they preferentially localize into sites of infection, namely the peritoneal cavity. The discussion provides some possible explanations for this puzzling finding (which is actually consistent with prior data), including that LANA itself inhibits antigen presentation and the presence of other APCs in vivo, like DCs, are necessary. Testing these ideas experimentally would have further strengthened the paper, but on its own, it provides important novel data. The following are suggestions for revision.

1. For C112, T2B35 cells were used, which is understandable given the antigen presentation for CD8 cells, while for all other clones, LCLs were used. It would be useful to describe the T2B35 line when first mentioned. Also, in many figures and figure legends (for example Fig 1, panel A), LCLs are described also for this clone. Unless I am misunderstanding, for C112 it should say T2B35 instead of LCLs, unless the authors say something like “antigen presenting cells (APCs)” throughout.

We appreciate the reviewer's observation and agree that clarification was needed regarding the use of T2B35 cells for C112. To address this, we have added a brief explanation in the main text on page 5 describing the T2 cell line transfected with HLA-B*35:01, along with the references to the original publications detailing the generation of T2B35 cells. We have also corrected the figure legend of Figure 1A/1C to indicate where CD8 C112 was stimulated with T2B35 cells. In all other figures, unless otherwise noted, we worked with HLA-matched allogeneic LCLs from different donors, and T2B35 cells were not used anymore.

2. On Line 298 referring to figure 5A, “mCD19 and mCB as markers for identifying the transgenic cells were not used for clustering and visualization”. While this makes sense, these markers were used in Figure 5C, so it would be useful to have an explanation for why they were excluded from the FlowSOM clustering (e.g. to annotate all the T cells regardless of transgene expression).

Thank you for this comment. We have clarified this point in the revised text on page 8 as follows: “mCD19 and mCB, which identify transgenic cells, were excluded from clustering and visualization analyses. This approach enabled the annotation of all T cells regardless of their transgene expression and allowed for the phenotypic projection of the TCR transgenic T cell population on this T cell heterogeneity.”

3. Figure 1, panel B. There are no labels on the two flow cytometry plots. The legend should define that the panel on the middle is untransduced and on the right transduced, or labels should be added to this effect on the figure.

We thank the reviewer for pointing this out and have revised Figure 1B to include appropriate labels, indicating the untransduced and TCR transduced samples.

4. Does the data suggest that in contrast to EBV, T-cell therapy for KSHV related diseases is unlikely to be successful? Albeit speculative, it might be nice to provide some thoughts on this in the discussion.

Thank you for this important question. As discussed in response to reviewer 1, comment 4, our data indicate that KSHV LANA-specific CD4 $^+$ T cells accumulate in response to viral load, likely driven by increased antigen exposure. We now provide additional experimental data that suggest that the TCR transgenic CD4 $^+$ T cells preferentially proliferate (new Figures 5G and S7E). Ki-67 expression was higher in TCR transgenic compared to untransduced T cells in all but one animal. This is now discussed on page 8 of the revised manuscript.

In line with this, Nalwoga et al. found that IFN γ responses to LANA (ORF73) were increased in individuals with detectable KSHV DNA in peripheral blood mononuclear cells, indicating an expansion of IFN γ producing cells in response to elevated viral loads. (Nalwoga, A. et al. *Nat Commun* **12**, 7323 (2021)). This observation aligns with findings in EBV infection, where T cell expansion can be driven by high viral loads, such as during infectious mononucleosis, yet may still exert protective effects in adoptive transfer settings in patients.

As for the protective capacity of these cells, our current data do not allow for definitive conclusions. As shown in Figure 4H, we did not observe a reduction in the frequency of infected animals following adoptive transfer of LANA-specific T cells. However, in other studies on protective T cell responses it was necessary to transfer 10^7 cells to achieve immune control of EBV transformed lymphoma cells (Mietz et al., *EBioMedicine* **106**, 105240 (2024); Palianina et al., *Sci Adv* **10**, eado2048 (2024)). Therefore, we consider the transfer of 2×10^5 TCR transduced T cells with which we obtained CD4 $^+$ T cell frequencies of 0.1% rather as indicator specificities which allow us to study the differentiation, tissue distribution and accumulation of distinct antigen-specific CD4 $^+$ and CD8 $^+$ T cell populations. In future studies we plan to compare EBV and KSHV specific T cell populations for their protective effect against double-infected B cells.

5. Very minor- perhaps replace the Cesarman 2011 reference for the review in Blood 2022 (PMID: 34479367) which is more up to date.

Thank you for this input. We adapted the reference accordingly.

Reviewer #3 (Remarks to the Author):

Böni et al. describes the cloning of KSHV LANA-specific TCRs and development of the TCRs transgenic mice. However, these T cells fail to recognize KSHV-infected B cells in vitro. The CD4 T cells are detected at infection sites in the humanized mice and develop into activated and early effector memory. These observations are interesting but remain preliminary. Numerous experiments can be carried out to enhance the study, which are detailed below.

Line 25: "T cells against KSHV, however, are barely detectable in infected individuals,": Most of these studies examined immunocompromised individuals, so this is not a fair statement.

Thank you for this consideration. While it is true that earlier studies often focused on immunocompromised individuals, more recent work has addressed this limitation. For example, Nalwoga et al. (*Nat Commun* **12**, 7323 (2021)) examined proteome-wide T cell responses against KSHV in HIV-negative individuals from KSHV endemic areas and found that responses were generally weak and heterogeneous. These findings are consistent with those reported by Roshan et al. (*Oncotarget* **8**, 109402 (2017)), which included participants from both, the FNLCR research donor program and the HIV and AIDS Malignancy branch clinic at the U.S. National Cancer Institute. These studies support the current wording, which reflects the limited and heterogeneous nature of KSHV-specific responses, even in immunocompetent individuals.

These studies are also referenced in the introduction on page 3 in the following paragraph: "Nevertheless, KSHV-specific T cell reactivities per individual appear to be limited, with patients recognizing on average only 1-5 different KSHV antigens, compared to a mean of 21 different EBV antigens (Nalwoga et al., 2021). Moreover, these KSHV-specific responses are highly heterogeneous in their viral antigen recognition between individuals (Robey et al., 2010; Roshan et al., 2014)."

Line 46 to 82: the comparison between EBV and KSHV is unfair because these studies had different types of populations: one predominantly in immunocompetent subjects while the other predominantly in immunocompromised subjects.

Thank you for this important point. We agree that differences in study populations, particularly regarding immune status, can influence the magnitude and breadth of virus-specific T cell responses. However, the comparison included in this section is based on studies that examined KSHV-specific responses in both immunocompromised and immunocompetent individuals. For example, Nalwoga et al. (*Nat Commun* **12**, 7323 (2021)) assessed HIV-negative individuals from endemic regions and still found that KSHV-specific T cells responses were generally weak, limited in breadth and in most individuals one magnitude lower in frequency of IFN γ producing cells. Similarly, Wang et al. (*Blood* **97**, 2366 (2001)) reported no detectable expansion of T cells following KSHV-seroconversion in HIV-negative individuals from the control group of the Pittsburgh portion of the Multicenter AIDS Cohort Study.

To address this concern of the reviewer and improve clarity, we made revisions to the text on page 3 to explicitly acknowledge the study populations and emphasize that the comparison is intended to illustrate general trends in T cell immunogenicity rather than implying a direct one-to-one comparison.

LANA is known to inhibit antigen presentation. Thus, the use of antigen needs to take this into consideration.

We acknowledge the concern about LANA's impact on antigen presentation and have discussed its immune evasion functions on page 10 in the discussion. We would like to clarify that the assays in our study address this concern as follows: Figure 2 assesses the functionality of transgenic T cells/the TCR using peptide-pulsed target cells, where synthetic peptides are directly loaded onto MHC molecules. This approach bypasses endogenous antigen processing and presentation, so LANA's known inhibition of antigen presentation does not affect this assay. In contrast, Figure 3 evaluates direct recognition of KSHV-infected B cells, which naturally process and present antigens despite possible LANA-mediated inhibition. Since B cells are natural reservoirs for KSHV and antigen-presenting cells, this experiment captures the biological relevance of antigen presentation under conditions where LANA may impair the pathway. Together, these figures address the concern by separating TCR functionality without antigen presentation (Figure 2) from recognition in the context of natural antigen processing and LANA's inhibitory effects (Figure 3). Nevertheless, LANA is in addition to K8.1 the most frequently recognized T cell antigen in HIV-negative individuals (Nalwoga et al., *Nat Commun* **12**, 7323 (2021)). Furthermore, we could observe that LANA specific CD4⁺ T cells accumulate in the peritoneum as the site of increased KSHV infected B cell persistence. Their activation as indicated by the EM1 differentiation of LANA specific CD4⁺ T cells might contribute to IFN γ production which primary immunodeficiencies (STAT4, IFN γ R1) have identified as protective against Kaposi sarcoma.

Line 190-197: PEL cells are usually derived from undifferentiated or minimally differentiated precursor B cells with minimal expression of cell surface markers. It is unclear how these LCL-derived cells could mimic the PEL cells. Many PEL cell lines have been isolated. The authors should test these lines.

In response to the comment regarding the origin of PEL cells and the use of LCLs in our study, we would like to clarify some aspects. The core gene expression signature of PEL cells has previously been described to resemble plasmablasts or plasma cells rather than minimally differentiated precursor B cells (Klein et al., *Blood* **101**, 4115 (2003)), as evidenced by their expression of activation and plasma cell differentiation markers, such as CD30, CD38, and CD138 or IRF4 along with the downregulation of common B cell surface markers like CD20, CD79a and surface immunoglobulins. These findings indicate that PEL cells are derived from a more mature B cell stage consistent with plasmablastic/plasma cell differentiation.

While PEL cell lines serve as important models of KSHV-associated malignancy, they also represent fully transformed cancer cells with complex mutational landscapes. In contrast, our co-infected LCLs, generated by infecting primary B cells from different HLA-typed donors with EBV and KSHV, allow us to investigate T cell recognition in the context of early infection. B cells are a known reservoir of KSHV *in vivo*, yet only a small minority of infected individuals progress to PEL. Thus, studying freshly co-infected B cells provides in our opinion a biologically relevant model for understanding how the immune system interacts with KSHV-infected B cells in the broader context of infection. Although LCLs are not identical to PEL cells, previous work by us and others (McHugh et al., *Cell Host & Microbe* **22**, 61 (2017); Hayes et al., *PLoS Pathog* **21**, e1013281 (2025)) analyzing LCLs derived from EBV and KSHV co-infected animals or after passaging of dually infected LCL cell lines in mice has shown that KSHV and EBV co-infected B cells share features of such plasmablast differentiation, including downregulation of CD86, ICAM1, and CD40 or upregulation of IRF4. These observations support the utility of EBV/KSHV co-infected LCLs as a biologically relevant model to explore immune recognition of KSHV in the context of EBV co-infection. We therefore revised our introduction on page 4 to emphasize the importance of EBV co-infection in PEL biology and KSHV persistence in B cells.

In response to the suggestion to test PEL cell lines directly, we have included new data (new Figure S4A) using the only HLA-matched PEL cell line known to us that expresses HLA DRB1*13. Our results corroborate previous findings reported by Sabbah et al. (*Blood* **119**, 2083 (2012)). Consistent with earlier studies on CD4⁺ T cell clones (Brander et al., *J Immunol* **165**, 2077 (2000); Nicol et al., *J Virol* **90**, 3849 (2016); Sabbah et al., *Blood* **119**, 2083 (2012); Shrestha et al., *PLoS Pathog* **17**, e1009091 (2021)), CD4⁺ T cells transgenic for KSHV specific TCRs exhibited poor recognition of unpulsed PEL cells, with significant responses observed only upon cognate peptide-pulsing. To reflect this, we propose adapting the sentence on page 6 to: "T cells, including the LANA-specific T cells clones whose TCRs were used in this study, have been shown to poorly recognize naturally infected PEL cells or KSHV-single-infected tonsillar B lymphocytes (Brander et al., 2000; Nicol et al., 2016; Sabbah et al., 2012; Shrestha et al., 2021). Furthermore, CI33 TCR-transduced T cells failed to recognize the HLA-matched HLA-DRB1*13:01 positive VG-1 PEL cell line unless peptide pulsed (**Fig. S4A**)."

In sum, while PEL cells as transformed cancer cells exhibit a plasmablastic/plasma cell phenotype, co-infected LCLs provide a complementary and valuable model for investigating KSHV infection and immune responses in B cells at an earlier and potentially more physiologically relevant stage of infection. Additionally, our testing against PEL cell lines confirms poor CD4⁺ T cell recognition unless exogenous peptides are provided, consistent with prior literature.

Figure 3D-E: unlike peptides, antigens from these cells need to be processed and presented in a proper way in order to be recognized by the T cells. Hence, one should not make a conclusion based on this *in vitro* assay.

We agree with the reviewer that, unlike synthetic peptides, antigens expressed by infected cells require proper endogenous processing and presentation by MHC molecules to be recognized by T cells. However, this complexity is precisely why we interpret the results of Figure 3D-E as evidence that B cells poorly present LANA to T cells. In these assays, T cells rely on the natural antigen-processing machinery within the infected B cells to generate and present LANA-derived epitopes. Robust T cell recognition would be expected only if B cells could efficiently process and present LANA peptides. The observed weak or poor recognition together with the results of Figure 2, showing robust recognition of peptide pulsed target cells with the correct HLA restriction elements, therefore suggests limited endogenous processing and limited presentation of LANA by B cells. We would also like to point out that our study has tested this *in vitro* in unprecedented breadth with MC116 B cell progenitors, Brk.219 B cells, EK LCLs, freshly co-infected B cells and the VG-1 PEL cell line.

Our results align with prior knowledge of LANA's immune evasion properties that we discuss on page 10 and provide meaningful insights into the biology of KSHV-infected B cells. Furthermore, they contrast with the LANA specific CD4⁺ T cell accumulation and effector memory differentiation in the peritoneum, the preferred anatomical site of double infected B cells in our humanized mouse model and one of the primary sites of PEL development.

Line 254-257: The authors use KSHV-infected mice reconstituted with human CD34⁺ cells to test the function of the isolated T cells. However, it is unclear what cells are infected? Are they latent or lytic? Do they develop LPDs? This model does not model PEL nor KS. The authors need to better characterize the status of infection and the impact on the infected cells.

We appreciate this concern but have characterized the KSHV infected cells in great detail in our previous publications on EBV and KSHV co-infection in humanized mice, including gene expression profiling, multiparametric flow cytometry, Flow-FISH, immune fluorescence histology and immunohistochemistry (McHugh *et al.*, *Cell Host & Microbe* **22**, 61 (2017); Caduff *et al.*, *Cell Rep* **35**, 109056 (2021); Caduff, Rieble *et al.*, *Nat Comm* **15**, 4841 (2024)). In a nutshell, we only found KSHV infection in B cells (CD20⁺IRF4⁻) and plasmablasts (CD20⁺IRF4⁺), 79% were co-infected with EBV by EBER Flow-FISH and double-infected cells displayed the core signature of PEL gene expression as defined by the Dalla-Favera and Carbone labs (Klein *et al.*, *Blood* **101**, 4115 (2003)). We have now better described these previous studies on page 4 in the introduction. Moreover, we provide additional experimental data by immunohistochemistry in the new Figure S6D that demonstrate that LANA expression can be found in CD19⁺ B but not CD3⁺ T cells in lymph nodes and spleen of KSHV and EBV double-infected humanized mice, further confirming restriction of KSHV infection to the B cell lineage in our mouse model. This has now been described on pages 7 and 8 of the revised manuscript text.

Together, these data support that the KSHV infected cells in our humanized mouse model are B cells that are in their majority co-infected with EBV, exhibiting at least in part PEL-like gene expression and a mixed latent and lytic KSHV gene expression similar to the AP3 PEL cell line (McHugh *et al.*, *Cell Host & Microbe* **22**, 61 (2017)). While this model does not fully recapitulate human PEL and lacks endothelial cell infection associated with Kaposi sarcoma, it provides a valuable platform to study KSHV infection of B cells and immune responses in a physiologically relevant context, particularly with respect to innate natural killer (NK) cell responses and adaptive T cell priming in response to viral persistence in EBV co-infected B cells.

The authors should try to confirm the presence of LANA-specific T cells in the peritoneal cavity of KSHV-infected human subjects as this observation is interesting.

We appreciate the interest in confirming the presence of LANA-specific T cells in the peritoneal cavity of KSHV-infected human subjects. Especially, given that the exact anatomical reservoirs of KSHV infection outside of peripheral blood, the tonsils or tumor tissues remain incompletely defined, and it is still unclear why PEL cells and infected B cells more broadly preferentially accumulate in body cavities such as the peritoneum. While extending immune monitoring to human body cavity sites would be valuable, such sampling in humans is challenging and often not feasible due to the rarity of PEL and ethical considerations. Therefore, our model provides an important complementary system to investigate LANA-specific T cell localization and function within relevant anatomical niches. We agree that future efforts could focus on these body cavity spaces in clinical settings where possible, especially also in light of the recent study by Hayes *et al.* (*PLoS Pathog* **21**, e1013281 (2025)) demonstrating the predominance of KSHV-infected B cells in the peritoneum after transfer of *in vitro* co-infected B cells into immune compromised mice. Thus, the humanized mouse model remains indispensable for exploring homing behavior of KSHV infected B cells and KSHV specific cell-mediated immunity at present.

Point-by-point response to the reviewers

We thank the three reviewers for their positive comments. We have now addressed the final concerns of reviewer #3, outline the respective changes in our responses below and highlight additions to the manuscript text by underlining.

Reviewer #1

Böni and colleagues report the isolation of several LANA-specific T cell receptors from KSHV LANA specific CD4 and CD8 T cell clones, their sequence and study their function after transfecting them into human primary T cells and into T cells from humanised mice. The main message of their manuscript is that these TCRs from CD4 and CD8 positive LANA-specific T cells only show low killing activity against KSHV-infected B cells and that CD4 T cells transduced with LANA-specific TCRs preferentially home to the peritoneal cavity (the site of primary effusion lymphoma) and differentiate towards an effector-memory phenotype. Their observations support the observation that KSHV, in contrast to EBV, only elicits a weak T cell response and suggest that KSHV-infected B cells are ineffective in presenting KSHV LANA antigens to CD4 and CD8 positive T cells. The unique contribution of this manuscript to the field of KSHV T cell immunity is that they could analyse the function of KSHV LANA-specific T cells in a small animal model of EBV/KSHV-coinfected mice. The manuscript therefore makes an important contribution to this field.

The authors have carefully addressed the three reviewers' comments, added further experimental detail to clarify a few technical points and clarified some of the unclear issues pointed out by the reviewers. I have no further suggestions.

We thank this reviewer for his/her supportive comments.

Reviewer #2

This manuscript has been revised, and prior reviewers' concerns have been addressed. I have no further comments or suggestions.

We thank this reviewer for his/her positive assessment of our study.

Reviewer #3

While I appreciate the authors' efforts to address several of my previous comments, I remain in disagreement with their responses to others, which I believe remain critical to the study's interpretation and impact. Although the authors cite a few studies that examined CTL responses in endemic KS, these studies are limited in scope, with small subject numbers and narrow approaches. The more appropriate and informative comparison would be to examine KSHV-infected subjects who do not develop malignancies, as this would provide a stronger framework for understanding immune control and pathogenesis.

We agree with this reviewer that many more studies on KSHV specific T cell responses should be conducted to obtain a more comprehensive picture of antigen dominance, magnitude and T cell phenotype. We are indeed trying to address such questions in the controlled setting of KSHV infection of a small animal model. However, the one study that we based the choice of interrogating LANA specific T cells in our animal model on and that reported the highest frequency for LANA and K8.1 specific T cell responses in KSHV seropositive individuals was conducted in 116 Ugandan individuals without KSHV pathologies and without HIV co-infection (Nalwoga et al., Nat Commun 2021). Therefore, the low KSHV specific T cell response and lack of antigen immunodominance seems to be characteristic for healthy KSHV carriers.

The above concern also directly impacts the choice of control groups (EBV). Without carefully selected controls, the conclusions regarding immune responses and viral interactions remain difficult to generalize.

We acknowledge that EBV co-infection could be a confounder in our studies. However, KSHV persistence in the absence of EBV is extremely rare and EBV co-infection has even been found as one of the strongest environmental risk factors for KSHV seropositivity (Sallah et al., Nat Commun 2020). Moreover, we have previously shown that in absence of the beneficial effects of EBV co-infection on KSHV persistence in our small animal model KSHV specific immune responses, such as a broad repertoire of KSHV specific antibodies, do not develop (Caduff, Rieble et al., Nat Commun 2024).

While up to 70% of PEL cases are co-infected with EBV, KS is uniquely associated with KSHV. This distinction must be acknowledged more clearly, as the introduction of EBV into the mouse model could complicate the interpretation of results.

We agree with this reviewer that we are only able to model one KSHV associated pathology in our small animal model, namely primary effusion lymphoma, which is in the majority of cases co-infected with EBV. We are not able to model KS and we have also no human endothelial cells in our mouse model from which KS is thought to emerge. In order to clarify this we have now added a sentence on page 4 to clarify that KS and KSHV associated multicentric Castleman's disease (MCD) do not harbor EBV co-infection in the tumor cells.

In summary, although the authors have strengthened the manuscript in some areas, these major concerns remain and should be directly acknowledged and discussed to improve the clarity and validity of the study's conclusions.

In order to address this concern of the reviewer, we have now acknowledged in the discussion that LANA specific T cells could still recognize tumor cells of KS and MCD, to further clarify that our findings mainly apply to double-infected PEL like tumor cells as the only KSHV associated pathology that we are currently able to model in small animals.